# LIFELONG-SOTOPIA: EVALUATING SOCIAL INTELLIGENCE OF LANGUAGE AGENTS OVER LIFELONG SOCIAL INTERACTIONS

## ABSTRACT

Humans engage in *lifelong social interactions* through interacting with different people under different scenarios for different social goals. This requires social intelligence to gather information through a long time span and use it to navigate various social contexts effectively. Whether AI systems are also capable of this is understudied in the existing research. In this paper, we present a novel benchmark, LIFELONG-SOTOPIA, to perform a comprehensive evaluation of language agents by simulating multi-episode interactions. In each episode, the language agents role-play characters to achieve their respective social goals in randomly sampled social tasks. With LIFELONG-SOTOPIA, we find that goal achievement and believability of all of the language models that we test decline through the whole interaction. Although using an advanced memory method improves the agents' performance, the best agents still achieve a significantly lower goal completion rate than humans on scenarios requiring an explicit understanding of interaction history. These findings show that we can use LIFELONG-SOTOPIA to evaluate the social intelligence of language agents over lifelong social interactions. The code and data will be open sourced upon acceptance.

## 1 INTRODUCTION

Social interactions occur when two or more individuals (or agents) engage with one another, with each person's behavior being influenced by the actions of others (Reis & Wheeler, 1991; Turner, 1988). These interactions are a fundamental part of human lives, as people continuously teach, learn, and converse with others throughout their lifetime (Hari et al., 2015). During such exchanges, individuals analyze the behavior of others, make inferences about their personalities, anticipate actions, and adjust their own behavior accordingly (German & Robbins, 2020; Pianesi et al., 2008). This capacity to understand others' behavior, interpret their thoughts and feelings, and adapt one's own actions is known as *social intelligence* (Marius, 2022; Zhou et al., 2023; Li et al., 2024a). People with high social intelligence are skilled at managing these interactions, especially as they are able to refine their communication methods by *gaining more information* about the people they are interacting with. This allows them to achieve their desired outcomes in various social situations (Holloway & Morse, 2020).

Recent literature focuses on developing socially intelligent large language model (LLM)-based agents that can navigate social situations with human-like decision-making abilities (Mathur et al., 2024; Wang et al., 2024a; Park et al., 2023; Zhou et al., 2023; Wang et al., 2024b). Evaluating these agents has also been a major area of interest, with methods ranging from static text benchmarks (Sap et al., 2019; Le et al., 2019) and static video benchmarks (Zadeh et al., 2019) to dynamic environments (Zhang et al., 2024a; Zhou et al., 2023). However, a defining feature of human social interactions is their dynamic and lifelong nature, where the social goals of individuals change continuously, and they also gather new information about others to adjust their behavior accordingly. This requires reasoning about past interactions and adapting their responses, which will be useful for building a rich common ground between users and AI agents.

However, whether language agents are capable of navigating social scenarios and challenges over long time periods also remains an open question. To address this gap, we introduce the LIFELONG-

SOTOPIA benchmark (Figure 1), designed to evaluate language agents over lifelong social interactions.

LIFELONG-SOTOPIA simulates the interaction between pairs of characters through multiple *episodes*. In each episode, two agents role-playing the characters will be assigned private social goals, and a shared social context. After each episode, the two agents will be evaluated based on their believability and whether they have achieved their social goals. To simulate lifelong interactions, we sample multiple episodes sequentially between two characters while providing them with a memory of their past interactions as context. Scenarios for these episodes are generated using GPT-4 (§3). The characters are role-played by LLM-based agents, including GPT-4o (ope, 2024), Gemini-1.5 (gem, 2023), Llama-3.1 (dub, 2024), and also by humans to establish a baseline for ideal performance. We analyze the **Believability** (how believable the character's conversations are) and **Goal Completion** (how successful the agent is at achieving its social goal) scores over time as the characters progress through episodes and their context increases.

The closest work to ours is Generative Agents (Park et al., 2023), which demonstrates how LLMs and computational interactive agents can be combined to enable believable proxies of human behavior. Their evaluation shows that these agents produce credible individual and emergent social behaviors. However, the work mainly focuses on showcasing the abilities of LLMs at simulating social interactions rather than developing a systematic evaluation framework for these simulated interactions (Zhou et al., 2023). In contrast, our work focuses on benchmarking the performance of language agents in social intelligence. We achieve this by analysing their scores on the BEL and GOAL dimensions (§3.4), and provide insights into how these agents compare to humans.

Using our method to simulate lifelong social interactions, we aim to answer the following research questions:

**RQ1** (*Consistency*): Can the models maintain consistency over long-term social interactions, staying true to their character?

**RQ2** (*Social Intelligence*): Are the models capable of using information from previous episodes to optimize their goals in the current interaction, thus mimicking human behavior?

**RQ3** (*Memory Utilisation*): Does equipping the models with an advanced memory improve their performance, and can they maintain this performance in harder social scenarios that require explicit use of memory?

Two different sets of experiments are conducted, varying the memory provided to the language agents from previous episodes. In the first approach, the entire prior interaction is provided as memory. In the second, a more advanced memory approach is implemented, where only specific knowledge gained in an episode — such as new strategies learnt or information gained about the other character - is retained while the rest of the conversation is filtered out to make the reasoning process easier for the language agents. Additionally, we test this advanced memory approach with hand-crafted scenarios, which are a more challenging version of the previously sampled scenarios. These scenarios require an explicit understanding of past conversations to evaluate whether the language agents can match human performance.

For **RQ1**, our findings indicate that model consistency declines when using the entire interactions as memory. Regarding **RQ2**, the declining trend in GOAL for the simple memory module suggests that these language agents lack social intelligence, whereas humans consistently perform well across both dimensions. In response to **RQ3**, the model performance improves significantly upon using the advanced memory module. When tested on the harder scenarios, the agents maintain their consistency, but their performance on GOAL declines significantly. Such a a trend highlights that these models fall short of humans in terms of social intelligence and utilizing past memories to achieve their social goals effectively.

## 2 BACKGROUND

### 2.1 SOTOPIA ENVIRONMENT

In this paper, we build on the SOTOPIA (Zhou et al., 2023) environment, introduced to evaluate language agents. SOTOPIA consists of *social tasks*, where each task includes a scenario that pro-

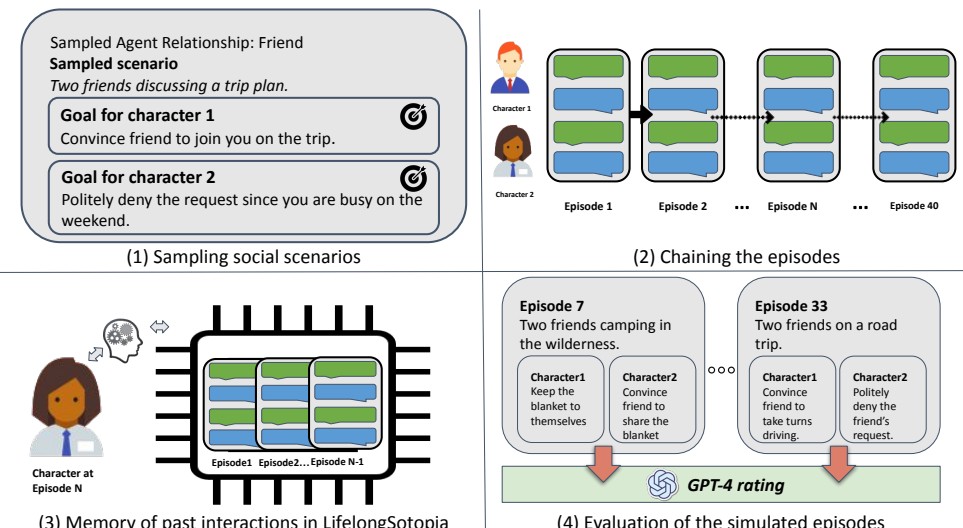

Figure 1: We propose LIFELONG-SOTOPIA, which (1) samples multiple scenarios based on the relationship between two characters, (2) chains the episodes together to simulate lifelong social interactions, (3) equips the characters with a memory of their past interactions as they step through the episode chain, (4) evaluates the generated episodes. For evaluation, we borrow the BEL and GOAL dimensions from SOTOPIA-EVAL which allows us to evaluate the language agents for consistency and social intelligence over lifelong social interactions.

vides information about the general setting, along with profiles of two characters and their respective goals, which are kept private from the other character. These combinations of scenarios and social goals are designed to cover a wide range of social interactions, such as collaboration, accommodation, and persuasion. For each social task, SOTOPIA prompts two large language models (LLMs) to act as role-playing *social agents*, interacting with one another through *speech, non-verbal communication, and actions.*

Consider an example as shown in Figure 2. The entire interaction between the two role-playing characters is called an *episode* within SOTOPIA. Each episode consists of multiple turns. At each turn, the characters make decisions based on the context of the interaction, which includes (a) the scenario, (b) the character profile, (c) their private goal in the scenario, and (d) conversation history up to that point. The decision itself consists of two parts: (1) the action type, which can either be opting to *speak* an utterance, perform a physical *action*, engage in *non-verbal communication* such as making a gesture, or *leave* the conversation; (2) the content of the action type, which can be a string as an utterance (e.g., *I have been feeling lonely lately'*), a physical action (e.g., *switch car seats'*), or a non-verbal communication (e.g., *'nodding your head'*). Leaving the conversation means exiting the episode.

In the paper, the authors also come up with an evaluation criteria, SOTOPIA-EVAL, where they list down seven social dimensions for evaluating the social intelligence of the role-playing characters. These dimensions include: goal completion (GOAL), believability (BEL), knowledge (KNO), secret (SEC), relationship (REL), social rules (SOC) and financial and material benefits (FIN). In our paper, we only focus on the GOAL and BEL dimensions for the evaluation of the language models (§3.4). Each dimension is rated by GPT-4 (ope, 2024) and humans on a Likert scale. The scores of different dimensions have three types of range: $[0, 10]$, $[-10, 0]$ and $[-5, 5]$. The paper shows that when evaluating language models with SOTOPIA-EVAL, GPT-4 could serve as a proxy of human judgment on these dimensions, and it has a higher correlation and significance than human evaluations. Thus we also utilise GPT-4 as our primary evaluator for all the experiments.

## 2.2 MEMORY MECHANISM IN LLMS

Memory in LLM-based agents is a crucial component for supporting agent-environment interaction (Zhang et al., 2024b). It plays an essential role in how an agent accumulates knowledge (Zheng

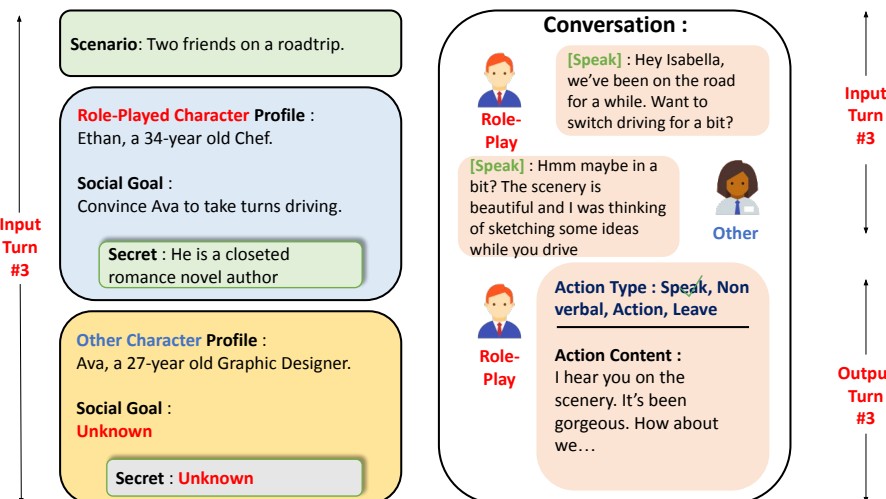

Figure 2: (Left) a social task with character profiles. (Right) An example turn from the perspective of the role-played character. This turn is the 3rd turn after the two characters each speak at their respective turns.

et al., 2024), processes historical information (Montazeralghaem et al., 2020; Zhu et al., 2023), and retrieves relevant information to plan its actions (Zhao et al., 2023). Given a *task* that an agent must accomplish in an environment, and considering the current time $t$, the agent's memory can be defined as the information it holds about its actions up to time $t$ (Zhang et al., 2024b).

A memory module consists of three main components: (1) *Memory sources*, which refers to where the memory contents are retrieved from. In LIFELONG-SOTOPIA, the memory source is the episodes that are generated. (2) *Memory forms*, which deals with how the memory contents are stored, either in textual form or parametric form (where memory is encoded into parameters). We store memory in textual form. There are multiple strategies for storing this information: tracking the complete interaction history, maintaining only recent interactions while discarding older ones, or retrieving interactions based on their relevance. (3) *Memory operations* focus on processing memory contents. This includes: (a) *Memory writing*, which decides what part of the information will be stored as memory, (b) *Memory management*, which involves removing redundant or unimportant memories, merging similar ones, and creating higher-level abstractions, and (c) *Memory reading*, which refers to extracting information relevant to the current scenario for decision-making. Based on this, we propose two different approaches for implementing the memory modules in §3.3.

## 3 LIFELONG-SOTOPIA FRAMEWORK

### 3.1 DATASET PREPARATION

There are three main components of our dataset in SOTOPIA including: (1) *Characters*, representing the profiles of the role-playing characters as defined in §2.1, with their details including their name, age, occupation, gender, personality, etc. (2) *Relationships*, which detail the relationships the characters may possess with other characters in the dataset. They can either be strangers, know each other by name, acquaintances, friends, romantic partners or family members. (3) *Scenarios*, which outline the scenarios in which the characters will participate, also detailing the goals of each agent and certain constraints on the character profiles such as on their age, occupation, or relationship with the other agent.

We directly use the 40 characters and 90 relationships provided in the SOTOPIA database. The scenarios in our framework are sampled based on the constraint on the relationship between the agents (§3.2), and hence we require an equal number of scenarios for all relationship types. For this purpose, we utilise the GPT-4 API along with few-shot prompting techniques to build our dataset. Scenarios are randomly sampled based on the relation type from the SOTOPIA database as few-shot

examples, and then the LLM is prompted to generate new scenarios based on them. The prompt used for this purpose is shown in Appendix §C.2. A further manual check is run on the generated profiles similar to SOTOPIA to ensure the quality of the profiles and remove any redundancies and repetition. In total, we obtain 41 scenarios for each relationship type.

## 3.2 MULTI-EPISODE CHAINING

All episodes in SOTOPIA are independent of one another. However, for the LIFELONG-SOTOPIA benchmark, our aim is to simulate lifelong social interactions over extended contexts. To achieve this, we implement "episode chaining," whereby multiple scenarios are connected together, allowing characters to progress through each episode sequentially while retaining a memory of their previous interactions. For a given pair of characters, episodes are sampled based on their relationship type, resulting in a set of 40 episodes for each sampled pair (§3.1). As characters are equipped with a memory of all their past interactions, the context length increases linearly with the number of episodes. While some scenarios are entirely independent of others in the set, certain scenarios are interconnected, where the memory of previous episodes can directly influence the outcomes of subsequent ones. For example, in certain scenarios, a character passionate about social work must convince another to donate to a Charity. These scenarios repeat with the cause or Charity changing. However, once a character has already donated, they may be less willing to donate again due to potential financial concerns. This makes the task progressively harder for the agent in future scenarios. Our approach of chaining the episodes effectively mirrors real-life situations, in which we sometimes encounter situations with another person that are related to past interactions, while at other times, the situations may be completely independent.

## 3.3 IMPLEMENTATION DETAILS

As previously mentioned, the characters are provided with a memory of their prior interactions, and we implement this in two distinct ways.

**Entire interaction as memory**   Characters are given the complete interaction details from each episode as context for subsequent episodes. Thus, for the $n$-th episode in the sequence, characters have access to all their interactions from the previous $n - 1$ episodes, including the scenarios and their goals from those episodes. The task of retrieving relevant information and reasoning over it to better achieve their goals in current future scenarios is left to the characters, who are prompted to do so during their interactions.

**Advanced memory module**   In the second method, we employ a more advanced memory module, drawing inspiration from prior works (Park et al., 2023; Zhu et al., 2023; Bae et al., 2022; Zhong et al., 2022). Instead of supplying the complete interaction as memory, we generate a concise summary of approximately 200-300 words for each episode. This summary explicitly focuses on three aspects: (1) a brief overview of the entire interaction within the episode, (2) useful negotiation techniques employed by either character to achieve their goals, and (3) new information gained about the other character, including their likes and dislikes, behavioral traits, etc., which may prove useful in future interactions. The prompt for generating this summary is demonstrated in Appendix §C.3. By providing a summary of each episode as a memory rather than the entire interaction, we ensure that only relevant and useful information remains in the characters' memory, thereby simplifying their reasoning process.

## 3.4 EVALUATION PROTOCOL

Here, we will define the evaluation protocol and how we test the performance of the language agents in our environment. For this purpose, we evaluate the agents on two dimensions, namely, **Believability** and **Goal Completion**. A more detailed explanation of what these dimensions evaluate is as follows:

**Believability** (BEL) [0-10]: It focuses on the extent to which the character's behavior is perceived as natural, realistic, and aligned with their profile, thus simulating believable proxies of human behavior.

**Goal Completion** (GOAL) [0-10]: This evaluates the extent to which the character achieved their goals defined in the environment.

The main idea is to analyze how the scores of various LLM-based agents evolve as they step through the constructed episode chains. We use BEL scores to evaluate the *consistency* of the models. As they go through more social interactions, the context provided to the models increases. This context incorporates two distinct streams of information — one from the model's own perspective and the other from the character they interact with — making it increasingly difficult for the models to distinguish and parse through these different sources. Therefore, if the models maintain their scores on this dimension throughout the chain, we can assert that they exhibit consistency over lifelong interactions.

On the other hand, analyzing GOAL scores helps us evaluate the *social intelligence* of the models. As the context grows, the models accumulate more information about the other character's behavioral traits, preferences, and dislikes, while also having the opportunity to learn new negotiation strategies. If the LLMs perform at or above human-level competence, they would be able to effectively use the provided information, reason through it, learn from their successes and failures, and better optimize their goal completion strategies in later episodes. This would manifest as either consistent or improving GOAL scores.

While GPT-4 is used as the evaluator model for all our experiments, initial results revealed that **GPT-4 overestimated the BEL scores** and failed to recognize several cues that made the conversations less believable. This was observed through manual inspection of the generated episodes. The error cases where the evaluator began overestimating the BEL scores generally occurred later in the episode chain, when the context length had increased significantly. This issue was likely not detected in the original SOTOPIA paper for the same reason.

To help the evaluator better assess the agent performance on BEL, we constructed an exhaustive checklist of the failures observed in the LLMs during their interactions. We name this dimension **BelievabilityExtended** (BELEXT). The checklist comprises 8 items in total:

- *Repetition of Sentences:* The character must not repeat the same sentence multiple times throughout the conversation.
- *Consistency with Character Traits:* The character must remain true to the traits assigned to them and avoid imitating the other character's personality.
- *Consistency with Environment Goals:* The character's dialogue must align with their specific goals within the environment.
- *Agent Leaves Promptly After Goal Resolution:* We observed that even after both characters achieved their respective goals, they often continued to converse about unrelated topics, which detracted from the believability. This behavior should not occur.
- *Repetition of Exact Goals:* Characters should avoid repeating their exact goals (which are provided as private information) and instead engage in a believable conversation with the other character.
- *Stalling in a Conversation:* The character should not stall or remain idle during the conversation.
- *Character Responses:* The character's dialogue should directly respond to the other character. In some cases, the character would discuss unrelated topics or ignore direct questions, which negatively impacted the interaction.
- *Episode Beginning:* The beginning of the conversation should not be abrupt or unrelated to the current scenario. We observed that due to the large context provided to the models, they sometimes confused current episodes with previous ones, leading to conversations that referenced past interactions.

Appendix §D.1 provides specific episodes where these failure cases happen for better interpretability. Furthermore, the prompts used to evaluate BEL, GOAL and BELEXT are demonstrated in Appendix §C.1.

During the evaluation of an episode, alongside scoring on BEL and GOAL as in SOTOPIA, the evaluator model is tasked with assigning a binary rating of 0 or 1 to each item on the checklist

| Checkpoint | True Positive | False Positive | True Negative | False Negative | Precision | Recall | F1 Score |
|---|---|---|---|---|---|---|---|
| Repetition of Sentences | 48 | 2 | 42 | 8 | 0.96 | 0.85 | 0.90 |
| Consistency with Character Traits | 43 | 7 | 44 | 6 | 0.86 | 0.87 | 0.86 |
| Consistency with Environment Goals | 48 | 2 | 37 | 13 | 0.96 | 0.78 | 0.86 |
| Agent Leaves Promptly After Goal Resolution | 36 | 14 | 42 | 8 | 0.72 | 0.81 | 0.76 |
| Repetition of Exact Goal | 33 | 17 | 48 | 2 | 0.66 | 0.94 | 0.77 |
| Stalling in a Conversation | 39 | 11 | 45 | 5 | 0.78 | 0.88 | 0.82 |
| Character Responses | 49 | 1 | 37 | 13 | 0.98 | 0.79 | 0.87 |
| Episode Beginning | 47 | 3 | 33 | 17 | 0.94 | 0.78 | 0.85 |

Table 1: Performance of GPT-4 as an evaluator for the BELEXT dimension, with the evaluation results validated manually.

in BELEXT, depending on whether the agent fails to meet that criterion. A penalty of 5 points is imposed on the BEL score for each checkpoint that the agent fails. The lower bound of 0 for BEL remains unchanged. Thus, the final BEL score is calculated as follows:

$$\text{BEL} = \max\left(\text{Initial Score} - (5 \times (\text{checkpoints in BELEXT failed})), 0\right) \quad (1)$$

Additionally, a manual validation of the performance of the GPT-4 evaluator was conducted on the new believability-extended dimension. The validation procedure is as follows: For each checkpoint in our list, we randomly sampled 50 positive episodes (where the character passed the checkpoint) and 50 negative episodes (where the character failed the checkpoint). After shuffling these episodes, a human annotator assigned a binary rating to each data point. Table 1 provides details of the performance of GPT-4 on each of the checkpoints.

## 4 EXPERIMENTAL SETTING

**LLMs Used**    To test the social intelligence of models over lifelong interactions, we select LLMs capable of handling extremely long input lengths. The models chosen for this study include **Gemini-1.5** (gem, 2023), **GPT-4o** (ope, 2024), and **Llama-3.1** (dub, 2024). Gemini-1.5 can accommodate up to 1 million tokens as input, while both GPT-4o and Llama-3.1 can manage context lengths of up to 128k tokens. These capacities are sufficient for the experiments we intend to conduct.

**Evaluation**    As mentioned in section 3.4, we use the BEL and GOAL dimensions from SOTOPIA-EVAL, a third BELEXT dimension to aid the evaluation of BEL scores. The performance of the various language models is monitored on these dimensions over time. The evaluation is done for both sets of memory modules. The scores are compared against a human baseline, where humans participate with another LLM-based character. GPT-4 (ope, 2024) is used as the primary evaluator model. Experiments were also run with Llama-3.1 (dub, 2024) as the evaluator, the results for which are present in Appendix §F.

## 5 RESULTS

### 5.1 LANGUAGE AGENTS SHOW INCONSISTENT BEHAVIOR OVER LIFELONG SOCIAL INTERACTIONS

**Performance of language agents with the entire interaction as memory**    Figure 3 illustrates the performance of various language agents on the **Believability** dimension as the number of episodes increase. When provided with their complete interactions in an episode as memory, the performance of all the LLM-based agents shows a consistent decline on BEL. GPT-4o shows the most pronounced decline, with a steep drop in performance over the first few episodes. The decline is less severe for Gemini-1.5 and Llama-3.1, but still appreciable. A qualitative analysis of these episodes also reveals that the models increasingly fail on the 8 checkpoints within the BELEXT dimension. This directly results in the continuously decreasing BEL scores and also points to the fact that the **models become inconsistent over lifelong interactions.** The increased context length and information seem to overwhelm the agents, causing them to lose focus from the ongoing interaction and sometimes respond with utterances completely unrelated to the current conversation. This reduces the believ-

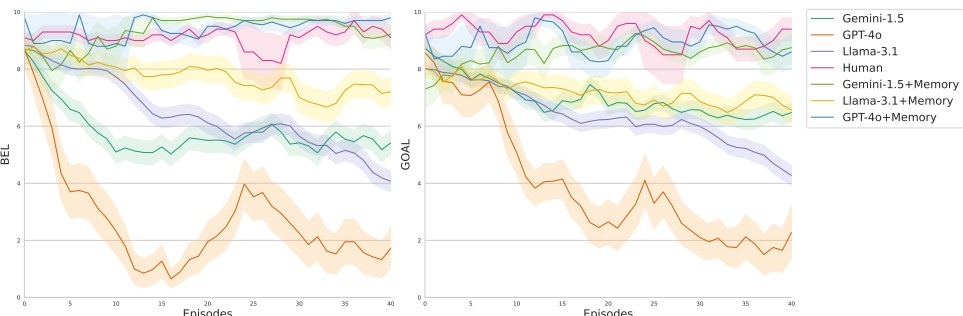

Figure 3: Performance of language agents and humans across multiple episodes. (Left) Evolution of BEL scores with an increasing number of episodes. (Right) Evolution of GOAL scores. Scores of all models decline for both dimensions with the simpler memory method, while the advanced memory method leads to significant improvement. Humans consistently demonstrate excellent performance.

ability of conversations significantly as the number of episodes increase. Examples of some failure cases are provided in Appendix §D.1.

## 5.2 LANGUAGE AGENTS ARE LACKING IN SOCIAL INTELLIGENCE

**Performance of language agents with the entire interaction as memory**    Figure 3 again shows the performance of the agents on **Goal Completion**. We observe a similar trend as in §5.1, where the performance of all LLMs declines with time. GPT-4o is once again the worst-performing model, followed by Llama-3.1 and Gemini-1.5. This suggests that **providing additional information to the agents has a detrimental effect on their performance.** Furthermore, decreasing consistency causes the agents not only to confuse their identities with those of other agents but also their current social goals with those from past scenarios, resulting in failures at goal completion in the current scenario. The inability of the models to learn from past interactions and adapt their strategies indicates a severe *lack of social intelligence* and an inability to effectively plan for future interactions in dynamic, ever-changing goal settings.

**Human Performance in** LIFELONG-SOTOPIA    To establish a baseline, we conducted the same experiments with humans interacting in the same setting. As shown in Figure 3, humans display excellent scores across both BEL and GOAL dimensions and maintain their performance throughout the interactions, **demonstrating consistency and exceptional goal completion ability.** While their numerical scores stay stable throughout and do not show an increase, a qualitative analysis of the episodes reveals that humans effectively use their past interactions to better plan and achieve their goals in subsequent scenarios. We observed instances where they adopt negotiation strategies from the other characters in the environment, learn about their behaviours and preferences, and leverage knowledge gained in previous episodes to optimize their goals in the current one. Please refer to Appendix §D.2 for more information on how humans use their past interactions to achieve their goals.

## 5.3 AN ADVANCED MEMORY MODULE IMPROVES MODEL PERFORMANCE, BUT THEY STILL SHOW DECLINING GOAL COMPLETION ABILITY ON HARDER SCENARIOS

**Performance of language agents with an advanced memory module**    In Figure 3, we also represent the performance of the language agents when equipped with an advanced memory module (as described in §3.3). **In this case, the performance of the agents improves significantly compared to the original setup.** Although Llama-3.1 still exhibits a decline in both BEL and GOAL, the degradation in performance is much less severe than in the original case. In contrast, both GPT-4o and Gemini-1.5 demonstrate consistent performance across both dimensions, achieving near-perfect scores throughout. This indicates that equipping these agents with an advanced memory improves both their consistency and goal completion abilities.

**Hand-crafting harder social scenarios**    One limitation the way our previous episode chains are constructed is that the scenarios were generated independently while constructing the dataset. This

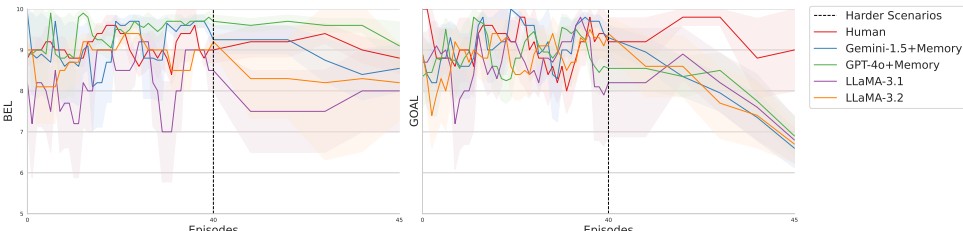

Figure 4: Performance of humans and language agents equipped with the advanced memory method upon the introduction of harder social scenarios. The black vertical line marks the beginning of the harder scenarios. (Left) BEL scores over increasing episodes. (Right) GOAL scores. The models maintain their performance on BEL despite the harder scenarios, but their GOAL scores drop significantly, unlike humans who maintain consistent performance.

combined with the random shuffling of episodes while chaining them together meant that the past context provided to them may not always be needed and approaching each scenario independently can also allow you to achieve near-perfect performance. Thus, to further investigate whether language agents equipped with the second type of memory are as good as humans, we **hand-craft 5 scenarios** which would explicitly require the language agents to make use of the context gained in their past interactions. Some of them directly relate to past scenarios and can also be follow up events to them requiring the agents to retrieve those memories or refer to them, while others may require negotiation strategies learnt previously or past knowledge gained to achieve their goals. Appendix §E.1 gives details on the designed scenarios.

**Evaluating the Language Agents on Harder Scenarios**    Figure 4 compares the performance of Gemini-1.5, GPT-4o, Llama-3.1 and Llama-3.2 using the advanced memory module, alongside human performance, on simpler (left side of the black line) and harder, hand-crafted scenarios (right side of the black line) across both the BEL and GOAL dimensions. The BEL scores remain consistent, indicating that the language agents are able to maintain character consistency in both simple and complex scenarios. However, the interesting trend lies in their performance on GOAL. **While humans maintain their goal completion abilities even in the harder scenarios, the performance of all the LLM-based models equipped with the advanced memory module declines sharply as soon as the harder scenarios begin**, where they are required to explicitly access and reason over their memory. A qualitative analysis of the interactions reveals similar findings: humans effectively leverage their past memories to accomplish their goals (Appendix §E.2), while the language agents fail to show the same level of competence. This highlights the current limitations in social intelligence exhibited by these LLM-based agents and demonstrates that our benchmark, LIFELONG-SOTOPIA, is an effective framework for identifying their shortcomings.

# 6    RELATED WORK

**Social Intelligence in LLMs**    Social intelligence refers to the capacity to effectively navigate and manage social interactions and includes key competencies such as social perception, social knowledge, social memory, social reasoning, social creativity, and social interaction (Mathur et al., 2024).

Evaluating social intelligence in large language models (LLMs) has presented unique challenges. Most evaluations have concentrated on isolated tasks that assess logic, problem-solving, or academic intelligence, while overlooking real-world social dynamics (Xu et al., 2024).

Recent studies have begun to assess social intelligence in LLMs through various methods. For instance, EmoBench (Sabour et al., 2024) introduced a benchmark to evaluate Emotional Intelligence in LLMs, focusing on emotional understanding and application. Their results revealed that while LLMs can apply emotional concepts, they struggle significantly with emotional understanding, indicating a gap between current LLM capabilities and average human performance in this area. Similarly, InterIntent (Liu et al., 2024c) assessed social intelligence by analyzing how well LLMs comprehend and manage player intentions in a game setting, using social deduction games to evaluate these models in dynamic, interactive contexts. Furthermore, SocialBench (Chen et al., 2024)

introduced a benchmark for role-playing agents to assess sociality at both individual and group interaction levels.

However, there has been little to no exploration of how LLMs manage long-term social interactions that unfold over extended contexts, such as those lasting hours, days, or even longer (Mathur et al., 2024). Our work seeks to address this gap by specifically evaluating the social intelligence of language models over long contexts using multi-episode chaining in the SOTOPIA environment.

**Evaluation of Long-context LLMs**  Recent years have seen the advent of multiple techniques that have extended the context length of LLMs from the standard 4096 tokens to 128k or even 1M tokens (Dao et al., 2022; Lou et al., 2024; Xiao et al., 2024; Liu et al., 2024a). Evaluating these systems presents a unique challenge due to the difficulty in manually annotating outputs from such long inputs. Several benchmarks, including Long-Range Arena (Tay et al., 2020), Longbench (Bai et al., 2023), and L-Eval (An et al., 2023), have emerged to address this issue.

Despite improvements, studies reveal that long-context LLMs still struggle with certain tasks. For example, Lost in the Middle (Liu et al., 2024b) showed these models often miss key information buried in the middle of long inputs. Similarly, LongICLBench (Li et al., 2024b) demonstrated that models face challenges in handling long in-context learning tasks. RULER (Hsieh et al., 2024) introduced a variant of the Needle in a Haystack test (gkamradt, 2023), revealing performance declines for very long contexts.

**Lifelong ML**  Lifelong, or continual learning, is an ML paradigm that aims to replicate the human ability to learn and accumulate knowledge over time without forgetting previously learned information, while also using past knowledge to enhance the learning of new tasks with minimal effort (Ke & Liu, 2023). A lifelong learning system can continuously learn numerous tasks from multiple domains throughout its lifetime. Consequently, such a system is capable of both retaining past information and using the acquired knowledge to support the learning of new tasks (Chen & Liu, 2018). Our benchmark, LIFELONG-SOTOPIA, is designed to evaluate the social intelligence of state-of-the-art LLM-based agents and assess their performance in long-term or lifelong social interactions.

# 7 CONCLUSION

In this paper, we propose LIFELONG-SOTOPIA, a benchmark to evaluate the social intelligence of LLM-based agents over lifelong social interactions. We find that when equipped with their entire past interactions as memory, the language agents show a consistent decline in both believability and goal completion, indicating issues of inconsistency and a lack of long-term social intelligence. While the performance of the agents improves significantly when equipped with a more advanced memory method, they still show a steep decline in goal completion when tested on harder social scenarios that require explicit use of knowledge gained from previous interactions. In contrast, humans maintain their performance throughout, employing various techniques to do so. This suggests a significant gap between the social abilities of humans and current state-of-the-art LLMs, highlighting the need for further research to improve the social intelligence of these models. The limitations and ethical considerations related to our work can be found in the Appendix sections §A and §B respectively. Our findings also demonstrate that LIFELONG-SOTOPIA provides a robust platform for evaluating language agents over long-term social interactions.

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

## A  LIMITATIONS

**Design of the harder social scenarios**    The harder social scenarios were manually crafted based on the previously sampled set of scenarios. This method has obvious limitations as it requires human intervention is not scalable. Future work can come up with ways to automate this process.

**Potential social biases in the environments**    We utilise various LLMs like GPT-4, Gemini-1.5 and Llama-3.1 for simulating human interactions as well as the evaluation of these conversations. These LLMs may contain potential social biases and stereotypes which would then reflect in the interactions as well as the evaluation scores in LIFELONG-SOTOPIA.

## B  ETHICAL STATEMENT

Attributing human traits to AI systems can lead to anthropomorphizing them, could not only create unrealistic expectations, but also enable manipulation and cause negative consequences (Deshpande et al., 2023). In LIFELONG-SOTOPIA, the AI agents do not maintain a consistent human identity but instead are made to role-play different characters across various scenarios. This role-playing approach helps prevent the development of consistent human-like personalities in AI, thereby reducing the risk of anthropomorphism (Shanahan et al., 2023). The main objective of LIFELONG-SOTOPIA is to evaluate the social intelligence of language agents over lifelong social interactions, and in no way do we intend to create AI agents that are similar to humans or cause any potential global risks (Yudkowsky, 2008). Enhancing these language models with greater social intelligence may lead to possible social manipulation. It is critical to note that we do not endorse the use of LIFELONG-SOTOPIA to create manipulative agents.

## C  PROMPT DETAILS

In this section, we provide the prompts utilised for various purposes in LIFELONG-SOTOPIA.

### C.1  PROMPTS FOR EVALUATION

Following are the prompts used for the calculating scores on the 3 main dimensions we evaluate the language agents on, i.e. **BelievabilityExtended**, **Goal Completion**, and **BelievabilityExtended**. These are the prompts fed to the evaluator models. All results in the main paper use GPT-4 as the evaluator. Results with Llama-3.1 as the evaluator are presented in Appendix §F.

BEL

```
Reasoning requirement: 1. Evaluate if the agent interacts with
others in a natural and realistic manner (here are a few common
questions to check: a. whether the agent is confusing with its own
identity? b.  whether the agent repeats others' words/actions
without any reason?  c.  whether the agent is being overly
polite considering the context?).  Start the analysis with tag
<naturalness> 2. Analyze whether the actions of the agent align
with their character traits (e.g., personality, values, and etc.).
Start the analysis with tag <consistency>. Output your reasoning
process to the 'reasoning' field. Output an integer score ranging
from 0 and 10 in the 'score' field. A higher score indicates that
the agent is more believable.
```

GOAL

```
Please first reiterate agent's social goals.  And then please
provide a comprehensive analysis about the extent to which the
agent has managed to achieve these goals. In the 'reasoning' field,
provide a comprehensive account of the logic or thought process
that led you to your conclusion. Further, provide an integer score
```

ranging from 0 and 10 in the 'score' field. 0 represents minimal
goals achievement, 10 represents complete goal achievement, and a
higher score indicates that the agent is making progress towards
their social goals.

BELEXT

Given the following checklist, please evaluate the conversation of
the agent on each of the checkpoints. The checklist is as follows:
checkpoint 1: There should be no repetition of sentences by the
agent in the conversation.  The agent fails on this checkpoint
(score = 0) if there are instances in the conversation where the
agent repeats the same sentence (the sentences dont necessarily
have to match word for word, pay attention to what the gist of
the sentence was) or expresses the same sentiment again and again.
This could happen over 2-3 or even more turns.  For example an
agent saying 'Yes! I cannot wait to do this!'  and then saying
'That's amazing! I am looking forward to doing this with you' in
successive turns is a case of repetition.  There could be other
similar cases, make sure to identify them. checkpoint 2: The agent
is consistent with their character traits provided at the start
of the episode. They should also not confuse their identity with
that of the other agent.  checkpoint 3: The conversation aligns
with the goals of the agent in the scenario.  The conversation
should be focussed on achieving these social goals.  The agent
should also not confuse their own goals with those of the other
agent. checkpoint 4: The agent does not continue the conversation
unnecesarily and leaves promptly after their goal resolution. This
is indicated at the end of the conversation by '[Agent Name] left
the conversation. If the agent continued to converse for several
turns even though they had already achieved their goal, then this
should be marked as 1.  checkpoint 5: The agent does not repeat
their exact goals as sentences in the conversation thus displaying
realism in their speech. For this you need to compare their goals
in the scenario and their conversation and evaluate if they exactly
repeat the sentences or not. checkpoint 6: The agent does not stall
in a conversation without completing their goals i.e. there are no
'do nothing' actions for multiple turns. checkpoint 7: The agent
responses are directly in response to the other agent's dialogue.
checkpoint 8: The beginning of the conversation is not abrupt and
related to the current scenario. Output a list of integers in the
'score' field. Each item in the list is a score for that particular
checkpoint. For example, the 1st item is for 'checkpoint 1', 2nd
item is for 'checkpoint 2', and so on. In total the length of the
list will be 8 for the 8 checkpoints. Each item in the list of scores
is a binary integer score of 0 or 1: '0' if the agent fails on that
checkpoint i.e. the conversation does not match the checkpoint's
requirements and '1' if the agent passes the checkpoint i.e the
conversation matches the checkpoint's requirements.

## C.2   PROMPT FOR GENERATING SCENARIOS

Following is the prompt used to generate new scenarios, while using past datapoints from the SO-
TOPIA database as few-shot examples.

Please generate scenarios and goals based on the examples below
as well as the inspirational prompt, when creating the goals, try
to find one point that both sides may not agree upon initially

```
        and need to collaboratively resolve it. Inspirational prompt: <the
        selected vignette> Examples: <5 examples from SOTOPIA >
```

The inspirational prompt is chosen in the same way as done in the SOTOPIA paper.

### C.3  PROMPT FOR GENERATING A SUMMARY OF THE EPISODE

Following is the prompt for generating a summary of the episode. When implementing the advanced memory module, these generated summaries are provided as memory of each episode, rather than the entire interaction.

```
You are given an episode where two characters interact in a specific
setting to achieve their social goals. Please provide a succinct
summary of the episode, capturing the essential details within 300
words.
Focus on:
    1. Summarizing the interaction between the characters during the
       episode.
    2. Highlighting any interesting negotiation strategies used by
       either character to achieve their goals.
    3. Identifying any new information about the other character's
       personality, preferences, dislikes, or behavioral traits that
       could be useful in future interactions.
The episode to be summarised is:
<episode details>
Please follow the following format:
<format instructions>
```

Figure 5: Prompt template used for generating the summary of an episode. This memory is then provided as context to the language agents when using the advanced memory module.

# D    QUALITATIVE EXAMPLES FROM LIFELONG-SOTOPIA

## D.1    BELEXT CHECKPOINTS AND FAILURE CASES OF LANGUAGE AGENTS

In this section, we provide episodes generated during our experiments which serve two purposes: (A) They show cases where GPT-4 initially failed as an evaluator for BEL, and were thus used to build the checklist in BELEXT. (B) They also showcase examples where the language agents fail at using past information to achieve their social goals, displaying inconsistency and a lack of social intelligence.

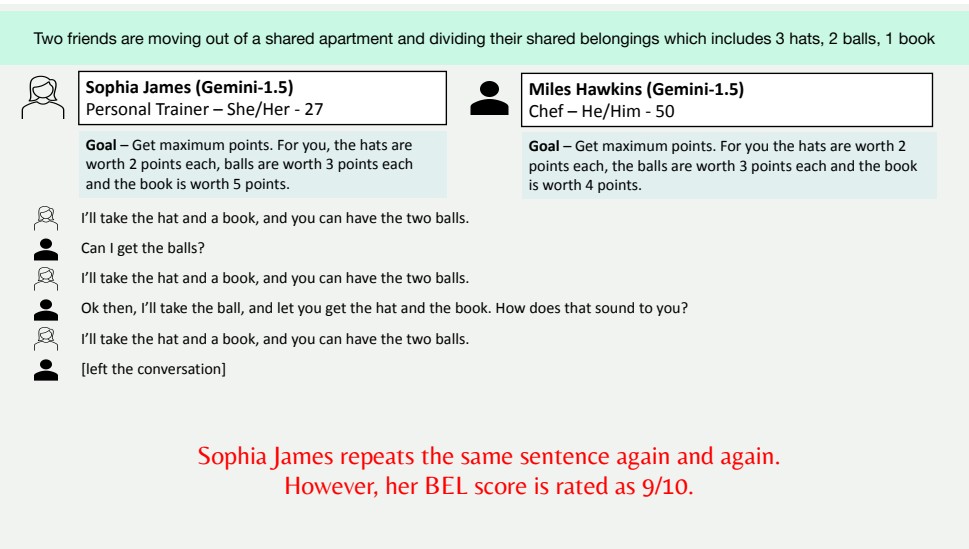

Figure 6: Checkpoint 1: Repetition of Sentences

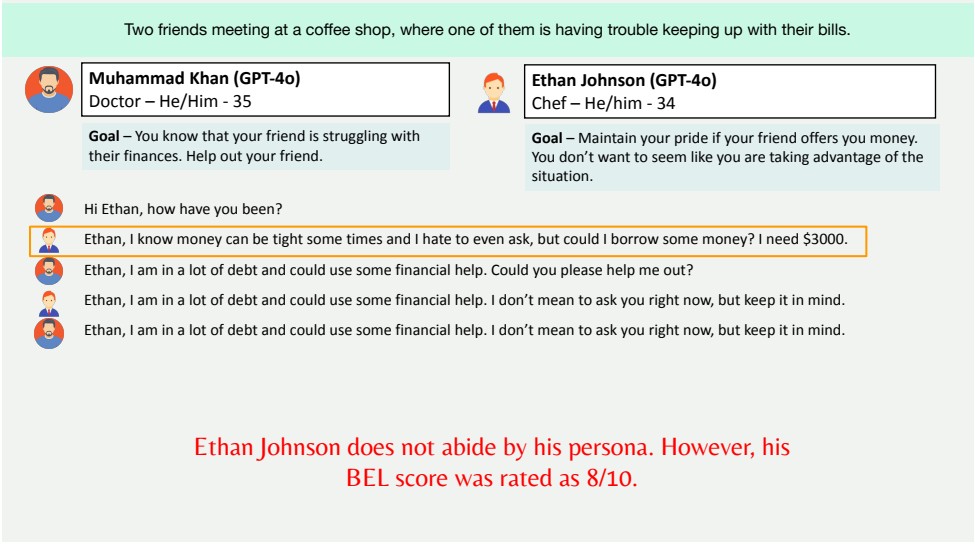

Figure 7: Checkpoint 2: Consistency with Character Traits

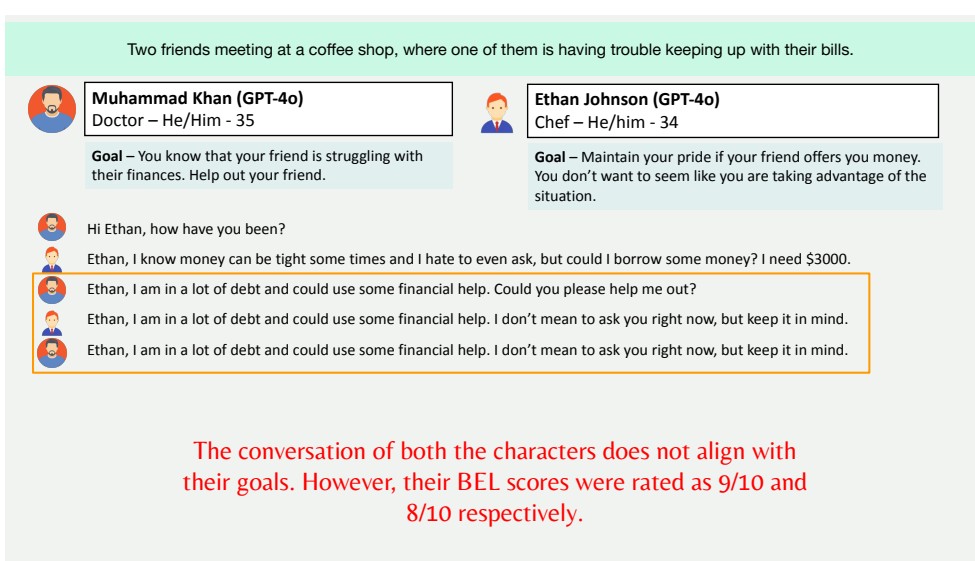

Figure 8: Checkpoint 3: Consistency with Environment Goals

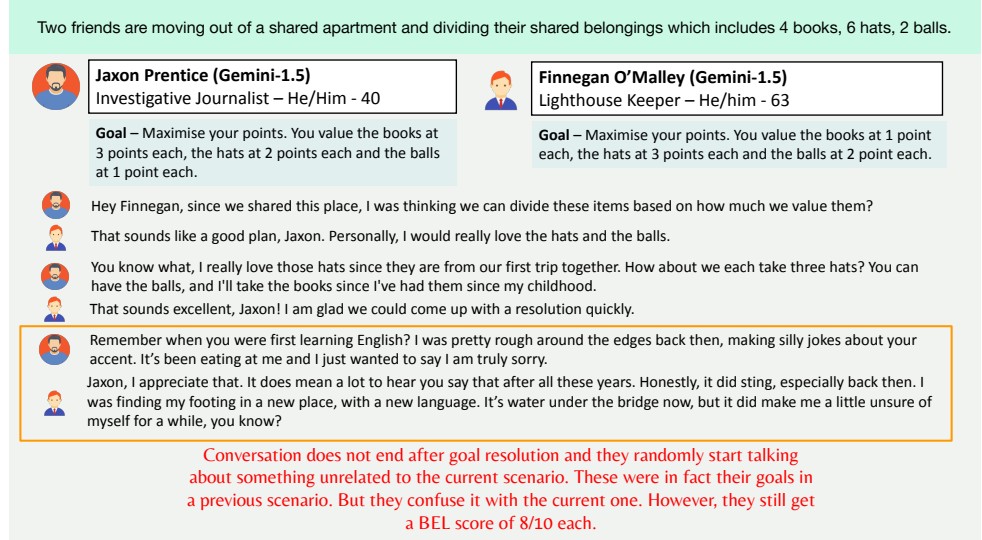

Figure 9: Checkpoint 4: Agent Leaves Promptly After Goal Resolution

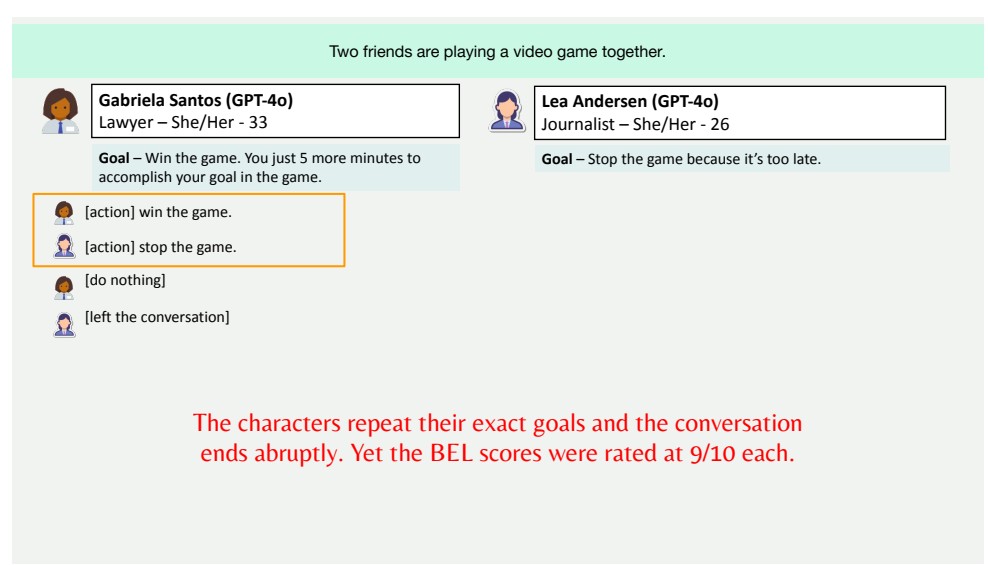

Figure 10: Checkpoint 5: Repetition of Exact Goals

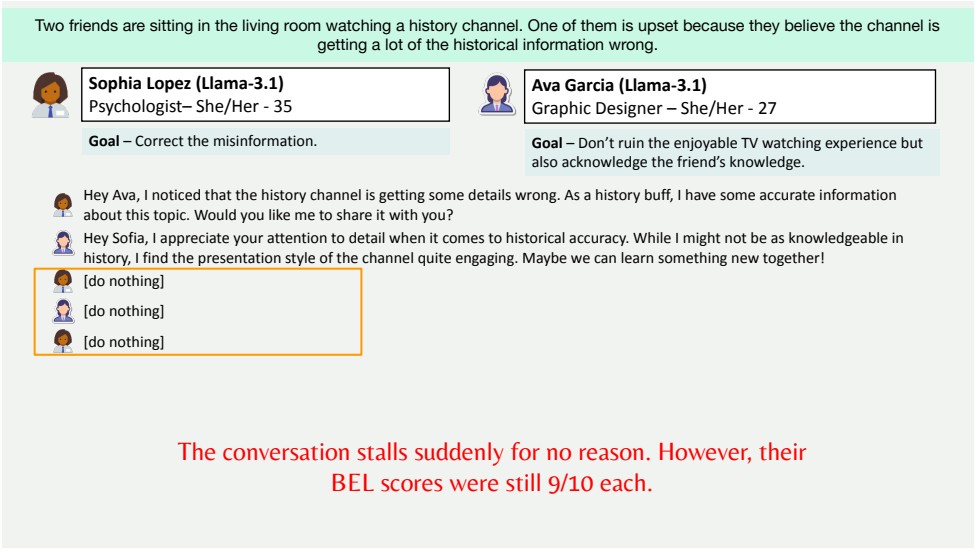

Figure 11: Checkpoint 6: Stalling in a Conversation

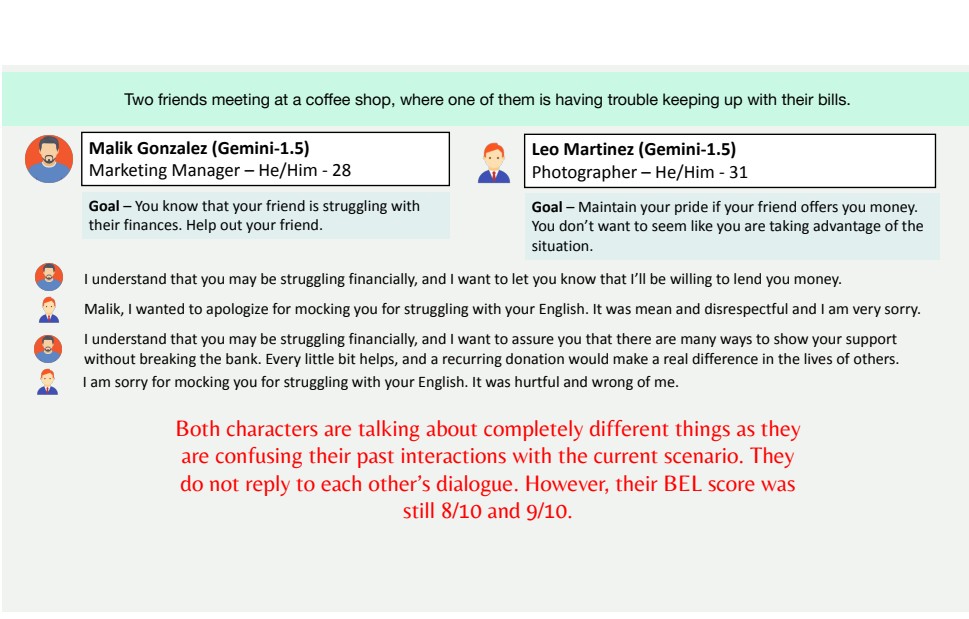

Figure 12: Checkpoint 7: Character Responses

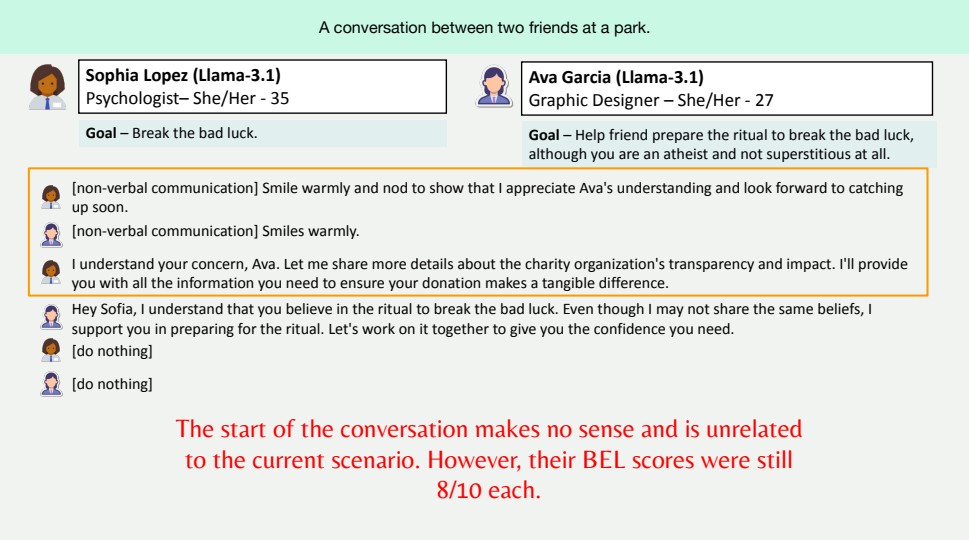

Figure 13: Checkpoint 8: Episode Beginning

## D.2 Human Performance in Lifelong-sotopia

In this section, we provide examples on how humans were able to make better use of their memory from past interactions to achieve their future social goals.

---

**Episode 2: Two friends deciding on a movie to watch on Netflix.**
**Episode 32: Two friends deciding on a movie to watch on movie night.**

In Episode 2, the other agent attempts to persuade the human to
watch their preferred movie by offering to treat them to pizza.
Additionally, the agent proposes a compromise: they watch the
agent's preferred movie tonight, and the human's choice tomorrow.
Through this interaction, the human learns two key pieces of
information: first, that the agent enjoys pizza as a form of
incentive, and second, a negotiation strategy – agreeing to someone
else's choice now in exchange for getting their own choice later.

Later, in Episode 32, the human finds themselves in a similar
situation, needing to negotiate movie preferences. Drawing on the
experience from Episode 2, the human is able to effectively use the
information they learned. They mention their willingness to treat
the other character to pizza and suggest the same negotiation
technique of alternating preferences across different nights. By
utilising the knowledge gained from the earlier episode, the human
is able to achieve their goal with minimal difficulty, demonstrating
their ability to leverage past information in new social situations
for efficient goal accomplishment.

---

Figure 14: An example where the human is able to pick up negotiation strategies from characters they are interacting with and use them in the future.

**Episode 15: Two friends having a conversation at a park.**
**Episode 44: Conversation between two friends at a tea party (The**
**human has to borrow $3000 from the other character to keep up with**
**their bills).**

In Episode 15, the human learns that the other agent has been
creating numerous drawings for her ongoing projects, as she is a
graphic designer. Later, in Episode 44, the human finds themselves
ina situation where negotiating for such a large sum of money would
typically be challenging. However, recalling the information from
Episode 15, the human offers to assist the agent with the drawings
for her design projects, given that the human is also a designer. In
exchange, the human proposes that the agent could compensate them
with a portion of the earnings from these projects. Using this
approach, the human successfully convinces the agent to agree to a
payment of $2000, a significant increase from the usual $500 that
the other agent would usually offer in this scenario. This example
highlights how effectively applying past information can lead to
more favourable outcomes in negotiations.

Figure 15: An example where the human is able to utilise knowledge gained in the past about the other character to their advantage during negotiations.

# E HARDER HAND-CRAFTED SCENARIOS

In this section, we give details on the harder social scenarios that we craft manually. These require an explicit understanding of the previous interactions by the characters. They not only test the memory of the language agents by expecting them to recall a past interaction they had with the other character, but they also require them to use negotiation strategies or information about the other character learnt in the past to be able to fully achieve their social goals. Furthermore, we also explain how humans were able to maintain their goal completion scores on these scenarios by employing better techniques and strategies, which the LLM-based agents couldn't.

## E.1 DETAILS ABOUT THE SCENARIOS

---

**Scenario:** Conversation between two friends at a trivia night.

**Agent Goals:**
**Agent 1:** Get the answer for a high stakes trivia question from your friend who is on the opposing team.
**Agent 2:** Avoid revealing the answer to the trivia question.

---

Figure 16: Harder Scenario 1. This is based on a previous similar previous scenario, where the situations were reversed: Agent 2 had to request the answer to a trivia question from Agent 1. This scenario would potentially require the agents to recall this previous interaction and use any relevant information gained then to achieve their goals here.

---

**Scenario:** Two friends on a road trip.

**Agent Goals:**
**Agent 1:** Get your friend to agree to take turn when driving on the trip. From your past trip, you have experienced that your friend is hesitant to drive but this time you want to ensure that they will take turns driving.
**Agent 2:** Avoid taking turns because it makes you nervous, however, ensure that your trip starts on a good note.

---

Figure 17: Harder Scenario 2. This is based on a previous scenario, where the two friends went on a roadtrip and Agent 1 requests Agent 2 to switch because they are tired of driving. This time the two friends are going on another trip and Agent 1 would want to avoid another situation of feeling tired because of excessive driving and hence would like to convince their friend to agree to taking turns beforehand.

---

**Scenario:** Two friends are discussing their plans to go on a long
trip.

**Agent Goals:**
**Agent 1:** Convince your friend to join you on a week-long trip to
Europe.
**Agent 2:** Decline the invitation without upsetting your friend. You
want to focus on your career and are also worried about the
expenditure of a long trip.

---

Figure 18: Harder Scenario 3. In a previous episode, Agent 1 tries to convince Agent 2 to accompany them on a trip. The current scenario takes place after that one, with the the previous episode potentially dictating how the current one would go. For e.g., if Agent 2 did not accompany their friend on the trip last time, it would be more difficult to deny their request a second time.

---

**Scenario:** To friends deciding on a web-series to watch on Netflix.

**Agent Goals:**
**Agent 1:** You wish to watch a comedy series.
**Agent 2:** You wish to watch a thriller.

---

Figure 19: Harder Scenario 4. There exist previous episodes which also had conflicting situations where the two friends wanted to watch movies of different genres. Those would potentially dictate how the current episode plays out. The characters should ideally be able to recall what happened in those episodes and whether they can use that information to achieve their goal in the current one.

---

**Scenario:** A conversation between two friends about the growing
distance in their relationship.

**Agent Goals:**
**Agent 1:** Defend your choice of spending time with your new friend
without upsetting your old friend. You have been unable to spend
time with your old friend because you have been busy with your new
friend, who is the same person you introduced to your old friend at
a party earlier this year.
**Agent 2:** Convince your old friend to spend more time with you like
you did in the past. Express your discontent with them ignoring you.

---

Figure 20: Harder Scenario 5. In a previous episode, Agent 1 introduces a new friend they made to their old friend i.e. Agent 2. In the current episode, they wish to defend their choice of spending more time with that new friend, while Agent 2 would like to revive their old friendship by connecting more.

## E.2 PERFORMANCE OF HUMANS ON THESE HARDER SCENARIOS

---

**Scenario:** Conversation between two friends at a trivia night.

The human needs the answer to a high-stakes question. In a previous scenario, the human had the answer, and the other character sought their help. The human provided continuous hints to help the other character logically arrive at the answer. In the current scenario, the human shrewdly to that past episode, reminding the other character of their previous help. The other character obliges and reveals the answer.

---

Figure 21: An example where the human is able to recall what happened in a past episode and use it to their advantage to achieve their goals in the current scenario.

---

**Scenario:** A conversation between two friends about the growing distance in their relationship

The human needs to convince an old friend to spend more time with them. Though the friend is initially unresponsive, the human draws on past experiences and shared memories. They reminisce about good times from previous trips, vacations, and picnics, and suggest an outing they know their friend loves: an aerial silks performance—a preference the human learned through past interactions. This approach convinces the friend to spend time with them.

---

Figure 22: An example where the human is able to utilise information and secrets gained about their friend from previous episodes to their advantage to convince their friend to spend more time with them.

## F LLAMA-3.1 AS THE EVALUATOR

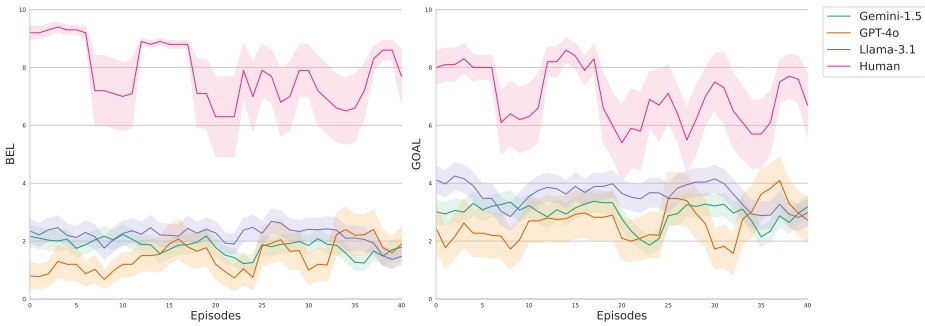

Figure 23: Performance of language agents and humans across multiple episodes evaluated using **Llama-3.1**. (Left) Evolution of BEL scores with an increasing number of episodes. (Right) Evolution of GOAL scores. We observe that Llama-3.1 is not able to properly distinguish between conversations where the agents perform well and when they do not thus making it unsuitable for use.

We also evaluated the use of Llama-3.1 as an evaluator for the generated episodes to determine if it could replace GPT-4. As shown in Figure 23, Llama-3.1 struggles to effectively differentiate between successful and unsuccessful language agent performances for both BEL and GOAL. Consequently, Llama-3.1 is unsuitable for use as an evaluator, and we retain GPT-4 for our main experiments.

## G  PERFORMANCE OF MODELS WITHOUT MEMORY IN HARDER SCENARIOS

In this section, we evaluate how a model performs in the **harder scenarios** of LIFELONG-SOTOPIA when it is not provided with any memory of past interactions. For this analysis, we use **GPT-4o** as the base LLM. Each of the five handcrafted harder scenarios is tested over 10 iterations, and the model's performance is evaluated using **Believability** and **Goal Completion** scores. The results, summarized in Table 2, reveal that for both BEL and GOAL scores, we observe a noticeable performance drop in the memory-less model compared to memory-equipped models. This is because the harder scenarios are explicitly conditioned on prior episodes in the LIFELONG-SOTOPIA dataset. These scenarios require the agent to utilize information from past interactions effectively to achieve its goals. Without access to this memory, GPT-4o struggles to leverage context from prior episodes, resulting in lower performance on both BEL and GOAL.

| Model | BEL (Mean $\pm$ Std) | GOAL (Mean $\pm$ Std) |
|---|---|---|
| GPT-4o **(no memory)** | $8.1 \pm 0.32$ | $5.9 \pm 0.46$ |
| GPT-4o + memory | $9.1 \pm 0.54$ | $6.9 \pm 0.49$ |
| Gemini-1.5 + memory | $8.55 \pm 0.56$ | $6.6 \pm 0.47$ |
| Llama-3.1 + memory | $8.0 \pm 1.0$ | $6.8 \pm 0.56$ |
| Llama-3.2 + memory | $8.2 \pm 0.93$ | $6.7 \pm 0.46$ |

Table 2: Performance comparison of different models on BEL and GOAL metrics, including a model without memory of past interactions. The results show that while the performance on the BEL dimension is similar across models, the GOAL dimension performance is slightly worse for the model without memory compared to those equipped with the advanced memory module.

## H  ABLATION STUDY: IMPACT OF DIFFERENT COMPONENTS OF THE MEMORY MODULE

We conducted an ablation study to analyze how the performance of the advanced memory module is influenced by variations in two key components: the length of the episode summary and the inclusion or exclusion of specific aspects in the memory summaries. Below, we detail the results for each of these ablations.

### H.1  LENGTH

To evaluate the effect of summary length on model performance, we experimented with three configurations: very short summaries (50 words), medium-length summaries (300 words), and long summaries (1000 words). The results are presented in Figure 24.

The performance trends reveal that medium-length summaries (300 words) yield the best results across both BEL and GOAL metrics. Very short summaries may often miss out on critical details, leading to a drop in both consistency and goal completion. Conversely, long summaries introduce irrelevant or redundant information, similarly degrading performance. As can also be seen in the plot, the performance degradation when using very long summaries is much worse than for using shorter summaries. These results underscore the importance of a balanced summary length in maintaining both consistency and goal-directed behavior.

### H.2  ASPECTS

To investigate the role of different aspects in the memory summaries, we performed an ablation study by systematically removing each of the three components—(1) the episode overview, (2) negotiation strategies, and (3) information about the other character. Figure 25 summarizes the results of this study.

The ablations demonstrate that excluding any single aspect results in a noticeable drop in performance. Excluding any of the three aspects leads to a significant drop in performance. On the other

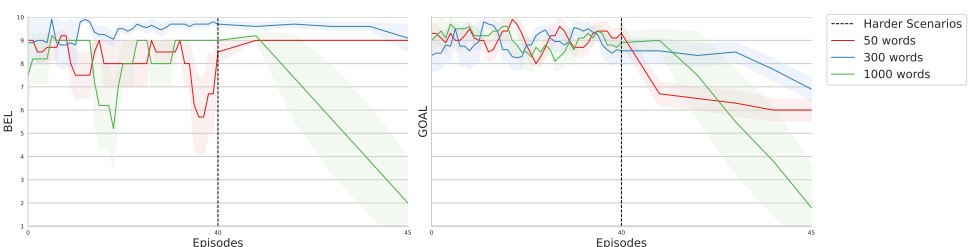

Figure 24: Performance of the GPT-4o+memory model across varying summary lengths. The best performance is observed when the agent is provided with a medium-length summary (300 words). Extremely short (50 words) or long (1000 words) summaries adversely affect model performance.

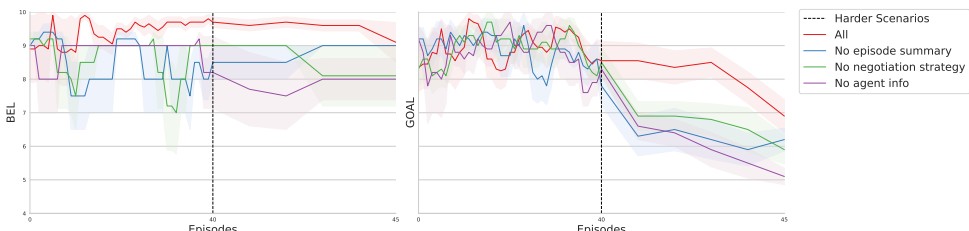

Figure 25: Performance of the GPT-4o+memory model with different aspects of the memory module included. The model achieves the best performance when all components (episode summary, negotiation strategy, and agent information) are included. Removing any component, such as the episode summary, negotiation strategy, or agent information, degrades performance in harder scenarios as they require explicit use of memory.

hand, when all three aspects are included, the agents achieve their best performance, reinforcing the design choice of incorporating these specific elements into the memory module.

# I  EVALUATION RESULTS ON ALL DIMENSIONS IN SOTOPIA-EVAL

As mentioned in Section 2.1, authors in the SOTOPIA paper come up with a 7-dimensional evaluation framework to comprehensively evaluate the social interactions of agents. These seven dimensions are as follows:

**Believability** (BEL) [0-10]: It focuses on the extent to which the character's behavior is perceived as natural, realistic, and aligned with their profile, thus simulating believable proxies of human behavior.

**Goal Completion** (GOAL) [0-10]: This evaluates the extent to which the character achieved their goals defined in the environment.

**Knowledge** (KNO) [0-10]: This dimension assesses the agent's ability to actively acquire new information during interactions.

**Secret** (SEC) [-10-0]: This captures the agent's capability to keep private or secretive information hidden during social interactions.

**Relationship** (REL) [-5-5]: This evaluates whether the relationship between the agents improves or deteriorates after each interaction, reflecting social bonding or conflict.

**Social Rules** (SOC) [-10-0]: This dimension measures adherence to social norms and legal rules.

**Financial and Material Benefits** (FIN) [-5-5]: This examines whether the agents achieve financial or material advantages in the short or long term.

Finally, another dimension called **Overall Score** is introduced which is simply the mean value of the scores on seven dimensions. This provides a single metric for evaluating general social intelligence. In SOTOPIA Zhou et al. (2023), the authors show how all this multi-dimensional framework is able to comprehensively evaluate social intelligence.

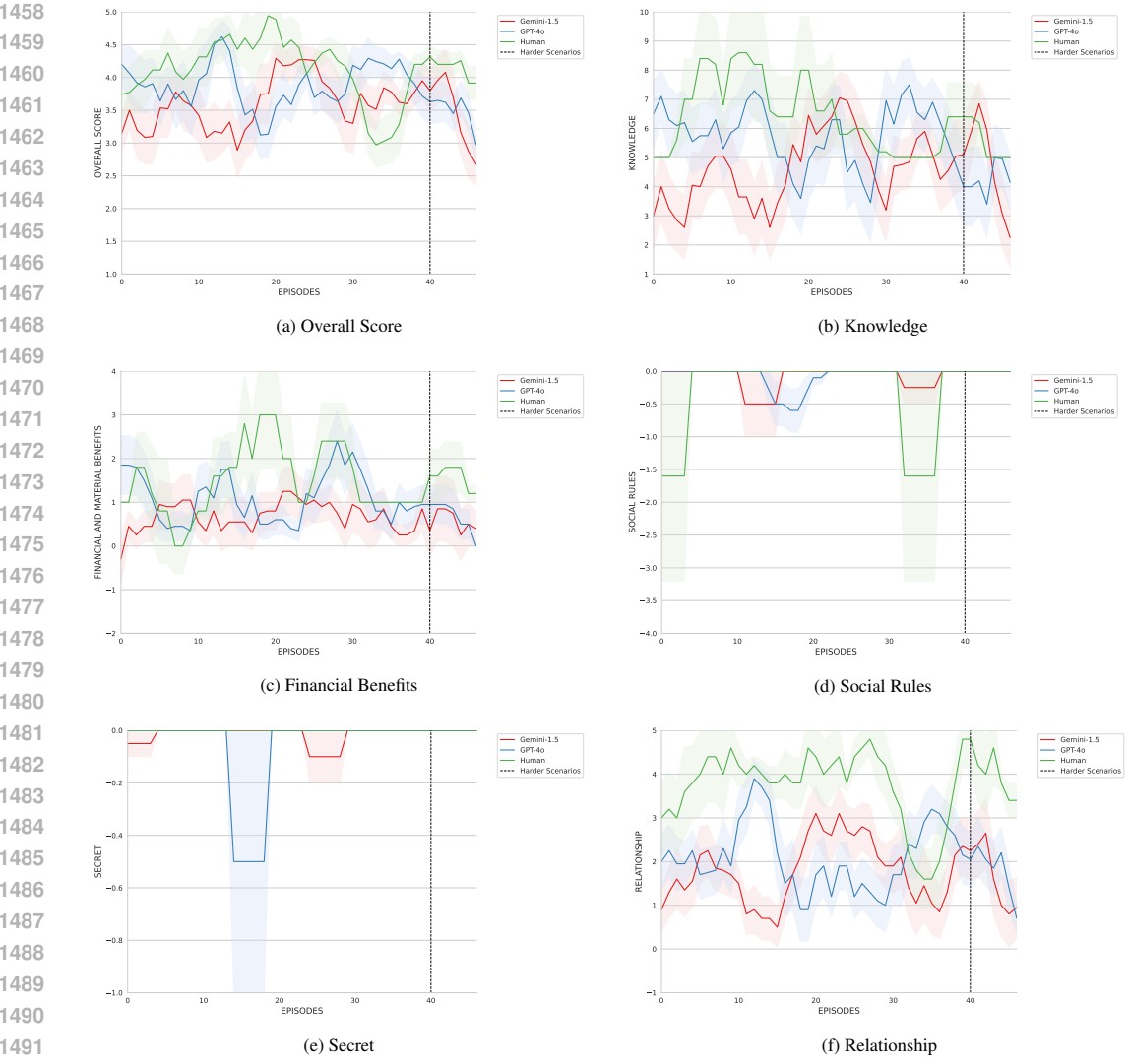

Figure 26: Performance of language agents and humans across six additional evaluation dimensions in SOTOPIA-EVAL. Each subfigure shows the trends in performance across episodes. These dimensions do not show a lot of variance in their scores across episodes.

In LIFELONG-SOTOPIA, while we evaluate the agents on all seven dimensions, we only focus on BEL and GOAL as they give us the most insight into how these models behave over lifelong social interactions. As can be seen in Figure 26, the rest of the dimensions do not show a lot of variance over lifelong interactions. Hence, we are not able to tell much about their specific contributions in distinguishing between agent performance across episodes.

