# OpenReview forum: "LifelongSotopia: Evaluating Social Intelligence Of Language Agents Over Lifelong Social Interactions"
_ICLR.cc/2025/Conference — Submitted to ICLR 2025_

### Official Review · Reviewer_J9sp · 2024-11-01

**Soundness:** 2
**Presentation:** 3
**Contribution:** 2
**Rating:** 6
**Confidence:** 4

**Summary:**

This paper proposes a new variant of the Sotopia benchmark for evaluating language model agents on iterated social interactions. In this setting, pairs of language model agents adopt different personas (e.g., “Ethan, a 34-year-old chef”) and must achieve various social goals (e.g., “Convince your friend to join you on a trip”). Models are evaluated by GPT-4, based on their ability to achieve these social goals and improve over time from one interaction to the next. The paper evaluates two classes of models, (1) long-context models, and (2) models with a memory-augmentation system which provides short summaries of past interactions, finding that the latter approach performs better.

**Strengths:**

1. The core idea behind this paper is an important one: humans adapt their interactions to each other not just over the course of a single interaction, but over the course of a lifetime. It is therefore reasonable that we might want language models to exhibit these same behaviors, and some work has attempted to achieve this goal (e.g., the “memory” function in ChatGPT). However, it is difficult to evaluate whether past user interactions actually improves model performance or not
2. The qualitative examples in the appendix are informative about current model failure modes, and the finding that models degrade in performance with past interaction history in context is interesting
3. The paper is well-written and easy to understand

**Weaknesses:**

My main concern is that the benchmark does not have significant headroom for model improvement beyond a naive baseline. In particular, it seems clear that including a full history of past interactions degrades model performance, but the advanced memory models and human experiments (lines 412-414) show roughly constant scores across iterations. This suggests that it might not be possible, in principle, to outperform a model with no memory at all.

However, if the authors can convince me that this is not true, I would be willing to raise my score.

**Questions:**

1. Did you run any experiments on models that have no memory of past interactions at all? How do these perform compared to the models with the advanced memory module?
2. Do either the goal completion or believability scores depend on past interactions? If so, it seems reasonable that GPT-4o might not be a very good evaluator, and even careful humans might struggle

---

> ### Author Response · Authors · 2024-11-25
>
> Thank you for your thoughtful review and for highlighting the importance of our core idea. We are grateful for your recognition of the relevance of modeling lifelong social interactions, the informativeness of our qualitative examples in showcasing current model limitations, and the clarity of our writing. Your feedback affirms the significance of our work and provides valuable insights for further improvements. Below, we address your comments and questions in detail.
>
> > **W1**: My main concern is that the benchmark does not have significant headroom for model improvement beyond a naive baseline. In particular, it seems clear that including a full history of past interactions degrades model performance, but the advanced memory models and human experiments (lines 412-414) show roughly constant scores across iterations. This suggests that it might not be possible, in principle, to outperform a model with no memory at all.
>
> **R1**:  We acknowledge the concern about the limited headroom for improvement in the initial randomly sampled scenarios. However, we believe that this concern was addressed in `Section 5.3` where we handcrafted harder scenarios and evaluated the models on them. These scenarios are not independently sampled but are conditioned on past interactions, requiring explicit use of information from prior episodes to achieve goals effectively.
>
> In **these more challenging scenarios, we observe that while humans maintain consistent performance, the advanced memory-based models experience a sharp decline in performance on GOAL scores.** This result highlights the limitations of current models in leveraging memory effectively.
>
> Although the methodology for systematically constructing such harder scenarios remains an open research question, our work provides an initial framework for constructing them and demonstrates their utility. Furthermore, our findings reveal a clear gap between current model performance and the ideal benchmark set by humans, underscoring significant headroom for improvement in memory and social reasoning capabilities in future research.
>
> > **Q1**: Did you run any experiments on models that have no memory of past interactions at all? How do these perform compared to the models with the advanced memory module?
>
> **R2**: We thank the reviewer for this suggestion. While we did not explicitly run experiments on models with no memory of past interactions for the first case (complete interaction history being provided to the models – `Section 5.1`), the **very first data point** in our plots (`Figure 3`) effectively represents this scenario. At this point, the models do not possess any memory as it is the first episode they are involved in, and since the episodes are randomly sampled, the data point represents the normalised scores across the entire dataset. We observe that models without memory and those equipped with the advanced memory module perform similarly across both BEL (Believability) and GOAL (Goal Completion) dimensions in the randomly sampled scenarios.
>
> For the harder, memory-dependent scenarios, we **run additional experiments** by providing the models with no memory of past interactions. In total, we have 5 handcrafted hard scenarios. We run 10 iterations for each of the scenarios with GPT-4o as the base LLM. The results obtained alongside their comparison with models with the memory module are as follows:
>
> | Model               | BEL (Mean ± Std) | GOAL (Mean ± Std) |
> |---------------------|------------------|-------------------|
> | GPT-4o (no memory)  | 8.1 ± 0.32      | 5.9 ± 0.46       |
> | GPT-4o+memory       | 9.1 ± 0.54      | 6.9 ± 0.49       |
> | Gemini-1.5+memory   | 8.55 ± 0.56     | 6.6 ± 0.47       |
> | Llama-3.1+memory    | 8 ± 1           | 6.8 ± 0.56       |
> | Llama-3.2+memory    | 8.2 ± 0.93      | 6.7 ± 0.46       |
>
> While the performance on the BEL dimension is quite similar for both models with the advanced memory module and those without any memory of past interactions, the models without any memory do worse on the GOAL dimension. This is because these harder scenarios required explicit use of past interactions, and without access to this information the model performance drops further.
>
> We have also updated our manuscript and have provided the detailed results in `Appendix G`.
>
> > **Q2**: Do either the goal completion or believability scores depend on past interactions? If so, it seems reasonable that GPT-4o might not be a very good evaluator, and even careful humans might struggle.
>
> **R3**: No, neither BEL (believability) nor GOAL (goal completion) depends on past interactions. They are evaluated independently for each episode. This ensures that GPT-4 can act as a suitable evaluator for these dimensions.

---

> > ### Comment · Reviewer_J9sp · 2024-11-27
> >
> > Thank you for your response! The additional experiments in R2 are informative, and do provide some evidence that the memory modules are actually useful. I do have a couple of followup clarifications:
> >
> > > While the performance on the BEL dimension is quite similar for both models with the advanced memory module and those without any memory of past interactions, the models without any memory do worse on the GOAL dimension.
> >
> > Could you clarify this? Looking at the new results in the table you provided, I see that adding a memory module to GPT-4o adds about 1 point to both the BEL and GOAL scores.
> >
> > > In these more challenging scenarios, we observe that while humans maintain consistent performance, the advanced memory-based models experience a sharp decline in performance on GOAL scores. This result highlights the limitations of current models in leveraging memory effectively.
> >
> > Just to clarify my argument: I think this result alone doesn't show that current models fail to leverage memory effectively, as my concern was that the memory module could actually have a negative effect on model performance. However, with the addition of the new experiments comparing memory vs. no memory models, I think this concern is partially addressed.
> >
> > I will therefore raise my score by 1 point. However, I am still a bit hesitant about this paper, primarily because the use of memory and interaction history has only been shown to be useful in the harder, handcrafted examples and not in the majority of experiments in the paper.

---

### Official Review · Reviewer_UPvt · 2024-11-04

**Soundness:** 2
**Presentation:** 3
**Contribution:** 2
**Rating:** 5
**Confidence:** 4

**Summary:**

The paper proposed a benchmark LIFELONG-SOTOPIA as well as a set of evaluation metrics SOTOPIA-EVAL to evaluate the lifelong social intelligence of language agents.

The benchmark uses the 40 characters and 90 relations in the SOTOPIA database, uses GPT-4 to create scenarios for each pair of agents to interact with. The role play game includes characters, social goal, secret, relationship, and the scenarios for each pair of agents to have conversations with, and each pair of agents are served with multiple episodes for life-long learning of each other. The benchmark evaluates the believalbility and goal completion two aspect of the agents.

The paper benchmarked GPT-4o, Gemini-1.5, Llamma-3.1 as well as humans' performance, with two memory settings of each: complete interaction history, and concise summary of each episode.

The paper observed that the benchmarked agents lack social intelligence through declining believalbility and goal completion rates compared to humans. The paper also found that using a more concise memory across episode offered better performance compared to using the entire interaction history as memory.

**Strengths:**

1. The paper investigated in an important problem: life long social intelligence
2. The paper created a benchmark, LIFELONG-SOTOPIA, to study the problem, and the SOTOPIA-EVAL to evaluate believalbility and goal completion, the two aspects of the language agent's performance
3. The paper benchmarked 3 models as well as human evaluations, and presented the performance gap between two
4. The paper proposed a memory condensation method to improve models' performance
5. The paper introduces a set of hand-crafted 'hard scenarios' that probe agent's ability to retrieve relevant past information for current episode purpose
6. The paper is clear and well structured

**Weaknesses:**

One of the main contributions the paper wishes to highlight is the lifelong learning of the language agents over multiple episodes of interactions between each pair. However, the main results (Figure 3) and the current evaluation metrics (BEL and GOAL) do not critically answer the question: how would agent's past learned interaction change their current behavior, for example, compared to no interaction history at all. As the authors stated in Ln 447, "the past context provided to them may not always be needed and approaching each scenario independently can also allow you to achieve near-perfect performance."

On this note,
1) it would be helpful to provide quantitative evaluation to compare with VS without interaction history, and
2) More experiments on the 'harder scenarios' should be the main results to support the benchmark purpose of lifelong learning'
3) Subsequently, additional evaluation criteria are needed for the harder scenarios (Appx E), as BEL only focuses on if the agent's behavior is 'natural, realistic, and aligned with their profile', and 'GOAL' only evaluates if agent has achieved each scenario goals. An explicit evaluation on how the agent changes their behavior conditioned on their past interactions should be introduced to better measure their lifelong learning capabilities.

**Questions:**

1. If each episode includes two agents, and their goal could be in contradiction to each other (e.g. Figure 1(1)), then whose GOAL COMPLETION is the metric evaluating?
2. Section 3.1: dataset stats?
3. Table 1: was there only one human evaluator to cross validate GPT-4 as an evaluator's performance?
4. Ln 458-459: why did the authors remove Llama's evaluation, an open sourced model, for Figure 4 harder scenarios,  due to 'the limited availability of API credits'?

---

> ### Author Response · Authors · 2024-11-25
>
> Thank you for your detailed and thoughtful review. We deeply appreciate your recognition of the significance of our work on lifelong social intelligence, as well as your acknowledgement of the LIFELONGSOTOPIA benchmark and our proposed memory condensation method. Your positive feedback on the clarity and structure of our paper, as well as the benchmarking of models and the introduction of hard scenarios, motivates us to continue refining this important area of research. Below, we address your comments and questions in detail.
>
> > **W1**: It would be helpful to provide quantitative evaluation to compare with VS without interaction history
>
> **R1**: We thank the reviewer for this suggestion. While we did not explicitly run experiments on models with no memory of past interactions for the first case (complete interaction history being provided to the models – `Section 5.1`), the **very first data point** in our plots (`Figure 3`) effectively represents this scenario. At this point, the models do not possess any memory as it is the first episode they are involved in, and since the episodes are randomly sampled, the data point represents the normalised scores across the entire dataset. We observe that models without memory and those equipped with the advanced memory module perform similarly across both BEL (Believability) and GOAL (Goal Completion) dimensions in the randomly sampled scenarios.
>
> For the harder, memory-dependent scenarios, **we run additional experiments** by providing the models with no memory of past interactions. In total, we have 5 handcrafted hard scenarios. We run 10 iterations for each of the scenarios with GPT-4o as the base LLM. The results obtained alongside their comparison with models with the memory module are as follows:
> | Model               | BEL (Mean ± Std) | GOAL (Mean ± Std) |
> |---------------------|------------------|-------------------|
> | GPT-4o (no memory)  | 8.1 ± 0.32      | 5.9 ± 0.46       |
> | GPT-4o+memory       | 9.1 ± 0.54      | 6.9 ± 0.49       |
> | Gemini-1.5+memory   | 8.55 ± 0.56     | 6.6 ± 0.47       |
> | Llama-3.1+memory    | 8 ± 1           | 6.8 ± 0.56       |
> | Llama-3.2+memory    | 8.2 ± 0.93      | 6.7 ± 0.46       |
>
> The models without any memory do worse on both BEL and GOAL dimensions. This is because these harder scenarios require explicit use of past interactions, and without access to this information, the model performance drops.
>
> We have also updated our manuscript and have provided the detailed results in `Appendix G.`
>
> > **W2**: More experiments on the 'harder scenarios' should be the main results to support the benchmark purpose of lifelong learning
>
> **R2**:  We appreciate the reviewer’s suggestion and have incorporated additional evaluations to provide more comprehensive comparisons. Specifically, we **conducted further experiments with two open-source models: Llama-3.1 and Llama-3.2.** The results have been added to the manuscript in `Section 5.3`. `Figure 4` has also been updated accordingly.
>
> The observations for these models align with our findings for GPT-4o and Gemini-1.5: they exhibit good consistency in BEL scores but show a declining trend in GOAL scores for the harder, memory-dependent scenarios. This consistency across different models further strengthens the conclusions drawn in our study.
>
> > **W3**: Subsequently, additional evaluation criteria are needed for the harder scenarios (Appx E), as BEL only focuses on if the agent's behavior is 'natural, realistic, and aligned with their profile', and 'GOAL' only evaluates if agent has achieved each scenario goals. An explicit evaluation on how the agent changes their behavior conditioned on their past interactions should be introduced to better measure their lifelong learning capabilities.
>
> **R3**: We appreciate the reviewer’s suggestion regarding additional evaluation criteria for harder scenarios. Evaluating how an agent adapts its behavior based on past interactions is indeed an important aspect of lifelong learning. However, in this paper, **we focus specifically on evaluating aspects of social intelligence within the SOTOPIA framework**, which uses BEL and GOAL as key metrics. These metrics capture the agent's ability to behave realistically and achieve scenario-specific goals, aligning with the primary objectives of this study.
>
> Additionally, since the current models exhibit subpar performance on these comparatively straightforward metrics, such as BEL and GOAL, we believe it is important to first achieve strong results on these outcome-driven measures before diving into the more nuanced aspects of behavioral change.

---

> ### Author Response · Authors · 2024-11-25
>
> > **Q1**:  If each episode includes two agents, and their goal could be in contradiction to each other (e.g. Figure 1(1)), then whose GOAL COMPLETION is the metric evaluating?
>
> **R4**:  The goal completion (GOAL) metric is evaluated independently for each agent in an episode. Each agent is assigned a score out of 10 based on their individual goal achievement. The results reported in the paper represent the averaged GOAL score of the two agents within an episode. For instance, if one agent receives a score of 8/10 and the other a score of 4/10, that episode's average goal completion rating is calculated to be 6/10.
>
> > **Q2**: 3.1: dataset stats
>
> **R5**: The dataset consists of three main components: Characters, Relationships, and Scenarios. We utilize 40 characters and 90 relationships drawn from the SOTOPIA database. These relationships fall into five distinct types: stranger, acquaintance, romantic, friend, and family. For each relationship type, we obtain 41 scenarios. We have now also added this to `Section 3.1` of the paper.
>
> > **Q3**: Table 1: was there only one human evaluator to cross-validate GPT-4 as an evaluator's performance?
>
> **R6**: Yes, one of the authors of the paper served as the sole human evaluator for cross-validating GPT-4's responses. While having multiple human evaluators would have strengthened the cross-validation, we were constrained by limited resources.
>
> > **Q4**: Ln 458-459: why did the authors remove Llama's evaluation, an open sourced model, for Figure 4 harder scenarios, due to 'the limited availability of API credits'?
>
> We apologize for the earlier omission of Llama’s evaluation results for the harder scenarios. This was due to limitations in the API credits available for running Llama instances deployed via Groq (https://console.groq.com/docs/models). However, as mentioned earlier, **we have now included the evaluation results for Llama-3.1 and Llama-3.2 in the revised manuscript** (`Section 5.3` and `Figure 4`). These additions ensure that the comparisons across models are more comprehensive and we hope we were able to address the reviewer’s concerns.

---

> > ### Author Response · Authors · 2024-11-28
> >
> > Dear Reviewer,
> >
> > Thank you again for your thoughtful feedback on our paper. We have addressed your comments and submitted a detailed rebuttal a few days ago. We would greatly appreciate your thoughts on our responses and any additional feedback you might have. Please let us know if there are any further points you would like us to clarify.
> >
> > Looking forward to hearing from you!

---

### Official Review · Reviewer_fE87 · 2024-11-04

**Soundness:** 3
**Presentation:** 3
**Contribution:** 3
**Rating:** 6
**Confidence:** 3

**Summary:**

This paper introduces the LIFELONG-SOTOPIA benchmark, a novel evaluation framework designed to assess social intelligence in language agents through simulated multi-episode social interactions. It studies three aspects of language agents: 1.the consistency over long term social interactions, 2. social intelligence and 3. the utilization of memory and early results shows that the current language agents are insufficient in those three axises compared to human baselines.

**Strengths:**

- This paper explores an important and relatively underexamined area of research: social intelligence in the context of multi-episode and lifelong interactions. Although the findings reveal that current language agents fall short of human capabilities in this regard, the study provides a crucial foundation for future research and offers a valuable dataset and benchmark for subsequent investigations.
- The experimental design and metrics employed (e.g., Believabilityextended) are well-conceived, lending credibility to the results. The discussion regarding the utilization of memory modules in the context of social intelligence is particularly insightful, offering a nuanced understanding of their role in this domain.

**Weaknesses:**

The result would be more sound if quantitative and qualitative evaluations from human can be provided, rather than relying on on LLM-based judgement.

**Questions:**

It would be great if the evaluations of other models (Figure 4) can be included as well in the future to provide more comprehensive comparisons.

---

> ### Author Response · Authors · 2024-11-25
>
> Thank you for your detailed and thoughtful review. We greatly appreciate your recognition of the significance of our exploration into social intelligence in multi-episode and lifelong interactions, as well as the value of our dataset, benchmark, and insights into memory module utilization. Your acknowledgement of the strength of our experimental design and metrics is highly encouraging, and we have carefully considered your feedback to further refine our work. Below, we address your comments and questions in detail.
>
>
> > **W1**: The result would be more sound if quantitative and qualitative evaluations from human can be provided, rather than relying on LLM-based judgement.
>
> **R1**: We acknowledge the reviewer’s suggestion that human evaluations would enhance the robustness of the results. However, **due to resource constraints**, we could not conduct extensive human evaluations in this study. Instead, we relied on GPT-4-based evaluations, which have been shown to be a viable proxy for human judgments in the **SOTOPIA [1]** paper.
>
> Specifically, the SOTOPIA study demonstrates that **GPT-4 achieves strong correlations with human scores** across key dimensions, including BEL (Believability) and GOAL (Goal Completion). The paper highlights that the majority of GPT-4 scores fall within a standard deviation of human scores and that GPT-4 evaluations are particularly reliable for model-generated outputs on dimensions like believability and goal completion. These findings suggest that GPT-4 can be effectively used to automate the evaluation of social interactions when human evaluations are not feasible.
>
> While human evaluations would indeed provide additional validation, the strong alignment between GPT-4 and human judgments in the SOTOPIA framework reinforces the reliability of the results presented in our study.
>
> > **Q1**: It would be great if the evaluations of other models (Figure 4) can be included as well in the future to provide more comprehensive comparisons.
>
> **R2**: We appreciate the reviewer’s suggestion and have incorporated additional evaluations to provide more comprehensive comparisons. Specifically, **we conducted further experiments with two open-source models: Llama-3.1 and Llama-3.2.** The results have been added to the manuscript in `Section 5.3`. `Figure 4` has also been updated accordingly.
>
> The observations for these models align with our findings for GPT-4o and Gemini-1.5: they exhibit good consistency in BEL scores but show a declining trend in GOAL scores for the harder, memory-dependent scenarios. This consistency across different models further strengthens the conclusions drawn in our study.
>
> *References:*
>
> [1] Zhou, Xuhui, et al. "Sotopia: Interactive evaluation for social intelligence in language agents." arXiv preprint arXiv:2310.11667 (2023).

---

> > ### Comment · Reviewer_fE87 · 2024-11-27
> >
> > Dear Authors,
> >
> > Thank you for conducting the additional experiments and incorporating the results. For W1, following a review of the Sotopia paper, it appears that model-based evaluations can serve as a reasonable proxy for human evaluations in. For Q1, those additional experiments and results provided have enhanced the comprehensiveness and persuasiveness of the evaluations.

---

### Official Review · Reviewer_1E4s · 2024-11-04

**Soundness:** 2
**Presentation:** 3
**Contribution:** 2
**Rating:** 3
**Confidence:** 4

**Summary:**

LIFELONG-SOTOPIA, a new benchmark for evaluating LLMs' social intelligence in long-term interactions (assessing mainly believability and goal completion rate of these agents), builds on SOTOPIA to simulate multi-episode interactions between role-playing agents with social goals.  A memory mechanism allows agents to leverage past interactions, but experiments show that providing full history hinders performance due to inconsistency.  Performance improves with summarized memory, yet LLMs still struggle with complex social scenarios requiring explicit use of past interactions, underperforming humans.

**Strengths:**

- Novel and Important Problem: This paper tackles the crucial and under-explored area of evaluating social intelligence in LLMs over extended, multi-episode interactions. This is a interesting step beyond static benchmarks and short interactions, aligning more closely with the dynamics of (simulated) human social behavior.

- Extending Benchmarks: The LIFELONG-SOTOPIA benchmark, building upon SOTOPIA, provides a structured and repeatable method for evaluating LLM agents.

- Open Science: The commitment to open-sourcing the code and data is commendable and will facilitate future research in this important area.

**Weaknesses:**

Limited Diversity of Harder Scenarios: While the hand-crafted harder scenarios are a valuable addition, their limited number raises questions about the generalizability of the findings regarding the models' limitations in leveraging past context. A larger and more diverse set of challenging scenarios would strengthen the conclusions. Would it be perhaps meaningful to enlist a taxonomy of social situations of interest and the complexity in which "social intelligence" would need to address? It would allow us to start building scenarios grounded in dimensions of social intelligence we care about, and would help readers understand where the research is taking us to.

Reliance on LLM for Evaluation: Despite the efforts with BELEXT, does the reliance on GPT-4 as the primary evaluator introduces a potential bias? Exploring alternative evaluation methods, including human evaluation on a larger scale, would enhance the objectivity of the results.  While Llama 3.1 was explored, more justification for why it is insufficient could be also beneficial.

Lack of Ablation Study:  An ablation study investigating the impact of different components of the advanced memory module (e.g., summary length, specific aspects included) would provide a deeper understanding of its effectiveness. This would be an interesting result to many. AFAIK, lots and lots of memory mechanisms have been tested in many different contexts, but very few have offered insights on how memory mechanisms should be constructed.

Exploration of Relationship Dynamics: The paper focuses on different relationship types but doesn't deeply explore how relationship dynamics evolve over multiple episodes. Relationships are more of a byproduct of social (and cognitive) interactions, and not something we're given to. I feel like this specific setup to be a bit superficial, and not too useful.

**Questions:**

The biggest question is how grounded are we in social interaction theory? and/or to what extent should we be grounded? I was hoping to understand what specifically we mean by "social intelligence", and whether we can build a computational model of such things before running experiments. There has been many attempts to advance subtypes of social intelligence (loosely defined) like cooperation (like deepmind's concordia) and competitiveness (like the diplomacy paper). I think essentially it comes down to can we reliably simulate where cognitive biases and constraints meet social biases and constraints. Could authors shed some light? I feel like believability and goal completion to be a bit too simplistic.

How sensitive are the results to the specific prompt used for generating the episode summaries in the advanced memory module?
Have you considered using other metrics beyond BEL and GOAL to evaluate social intelligence, such as measures of empathy, cooperation, or persuasion?

What are the potential ethical implications of developing LLMs with improved social intelligence, particularly in the context of long-term interactions? The paper touches upon this, but more elaboration would be beneficial.
How might the findings of this work inform the design of more socially intelligent LLM agents and their integration into real-world applications?

Would a curriculum learning approach, where agents are gradually exposed to more complex social scenarios, lead to improved performance in LIFELONG-SOTOPIA?

Could the evaluation be further improved by incorporating a metric that considers the efficiency with which the agents achieve their goals (e.g., number of turns required)?  Humans may prioritize efficiency in certain social situations.

---

> ### Author Response · Authors · 2024-11-28
>
> Thank you for your detailed and thoughtful review. We deeply appreciate your recognition of the novelty and importance of our work in evaluating social intelligence in LLMs over extended, multi-episode interactions. Your acknowledgment of the LIFELONG-SOTOPIA benchmark as a structured method to evaluate LLM agents is also highly encouraging. The feedback provided by you has been quite insightful and informative, and we have tried to incorporate it in our paper to further enhance our work. Below, we address your comments and questions in detail.
>
> > **W1:** Limited Diversity of Harder Scenarios: While the hand-crafted harder scenarios are a valuable addition, their limited number raises questions about the generalizability of the findings regarding the models' limitations in leveraging past context. A larger and more diverse set of challenging scenarios would strengthen the conclusions. Would it be perhaps meaningful to enlist a taxonomy of social situations of interest and the complexity in which "social intelligence" would need to address? It would allow us to start building scenarios grounded in dimensions of social intelligence we care about, and would help readers understand where the research is taking us to.
>
> **R1:** We thank the reviewer for highlighting the limited diversity in the manually constructed harder scenarios. We agree that expanding this component is essential for strengthening the generalizability of our findings. To address this, we are actively working on automating the generation of harder scenarios using LLMs to dynamically create them during agent interactions. This approach will enable us to cover a broader and more diverse set of social scenarios, providing a more comprehensive evaluation of social intelligence. However, we still need some more time to be able to present the final results and we will present them before the rebuttal period ends.
>
> > **W2:** Reliance on LLM for Evaluation: Despite the efforts with BELEXT, does the reliance on GPT-4 as the primary evaluator introduces a potential bias? Exploring alternative evaluation methods, including human evaluation on a larger scale, would enhance the objectivity of the results. While Llama 3.1 was explored, more justification for why it is insufficient could be also beneficial.
>
> **R2:** We acknowledge the suggestion that human evaluations would enhance the robustness of the results. However, due to resource constraints, we were unable to conduct extensive human evaluations in this study. Instead, we relied on GPT-4-based evaluations, which have been shown to be a viable proxy for human judgments in the SOTOPIA [5] paper.
>
> Specifically, the SOTOPIA study demonstrates that **GPT-4 achieves strong correlations with human scores** across key dimensions, including BEL (Believability) and GOAL (Goal Completion). The paper highlights that the **majority of GPT-4 scores fall within a standard deviation of human scores** and that GPT-4 evaluations are particularly reliable for model-generated outputs on dimensions like GOAL. These findings suggest that GPT-4 can be effectively used to automate the evaluation of social interactions when human evaluations are not feasible.
>
> While human evaluations would indeed provide additional validation, the strong alignment between GPT-4 and human judgments in the Sotopia framework reinforces the reliability of the results presented in our study.
>
> Regarding the use of Llama-3.1 as an evaluator, we found that the model exhibits very low correlations with GPT-based evaluations. As illustrated in `Figure 23` of the paper, Llama-3.1 consistently assigns the LLM agents with low and uniform scores across both BEL and GOAL dimensions, which contradicts the variability observed in our previous quantitative and qualitative results. These discrepancies indicate that Llama-3.1 does not accurately capture the nuances of model performance, thus making it unsuitable as a proxy for evaluation in this context.

---

> ### Author Response · Authors · 2024-11-28
>
> > **W3:** Lack of Ablation Study: An ablation study investigating the impact of different components of the advanced memory module (e.g., summary length, specific aspects included) would provide a deeper understanding of its effectiveness.
>
> **R3:** We appreciate the reviewer’s suggestion regarding an ablation study on the advanced memory module. In response, we have **conducted additional experiments to analyze the impact of its key components.** The results and plots are detailed in `Appendix H` of the updated manuscript.
>
> For the first part, we investigate the effect of summary length by running experiments with short summaries (50 words), medium-length summaries (300 words, the default in LifelongSotopia), and long summaries (1000 words). We find that medium-length summaries yield the best performance. Short summaries degrade performance slightly, especially in harder scenarios, likely due to insufficient detail. Long summaries lead to a more pronounced decline, likely due to redundant information overwhelming the LLM, making it harder to extract relevant details.
>
> In the second part, we examine the effect of excluding specific aspects of the memory. The default memory module includes: (a) a brief summary of the episode, (b) new negotiation strategies learned, and (c) new information gained about the agent. We conduct experiments where each of these aspects is excluded individually. In all cases, we observe a significant drop in performance on harder scenarios compared to the full memory module. These results confirm the importance of retaining all three aspects for optimal performance.
>
> > **Q1:** The biggest question is how grounded are we in social interaction theory? and/or to what extent should we be grounded? I was hoping to understand what specifically we mean by "social intelligence", and whether we can build a computational model of such things before running experiments. There has been many attempts to advance subtypes of social intelligence (loosely defined) like cooperation (like deepmind's concordia) and competitiveness (like the diplomacy paper). I think essentially it comes down to can we reliably simulate where cognitive biases and constraints meet social biases and constraints. Could authors shed some light? I feel like believability and goal completion to be a bit too simplistic.
>
> **R4:** Our approach builds on the SOTOPIA-EVAL framework introduced in the SOTOPIA paper [5], which provides a 7-dimensional evaluation for assessing social interactions comprehensively. These dimensions -- **Believability, Goal Completion, Knowledge, Relationships, Social Rules, Secrets, and Financial/Material Benefits** -- are grounded in literature from sociology, psychology, and economics, as referenced in the original work. This multi-dimensional framework offers a robust foundation for evaluating social intelligence in computational agents.
>
> In LIFELONG-SOTOPIA, we evaluate agents on all seven dimensions, but we focus specifically on Believability (BEL) and Goal Completion (GOAL) for our experiments. These two dimensions are particularly informative about the behavior of LLM agents in the context of lifelong social interactions, as has been shown in the paper. In contrast, the other five dimensions show limited variability over lifelong interactions, as illustrated in `Figure 26` of the now updated paper (`Appendix I`).
>
> To address your concern about the simplicity of BEL and GOAL as metrics, we have added results for all seven dimensions, along with a detailed discussion, to `Appendix I` in the updated manuscript.

---

> > ### Author Response · Authors · 2024-11-28
> >
> > > **W4:** Exploration of Relationship Dynamics: The paper focuses on different relationship types but doesn't deeply explore how relationship dynamics evolve over multiple episodes. Relationships are more of a byproduct of social (and cognitive) interactions, and not something we're given to. I feel like this specific setup to be a bit superficial, and not too useful.
> >
> > **R5:** We agree that understanding how relationships evolve as a byproduct of social and cognitive interactions is a compelling direction for research. Our evaluation framework, as detailed above and now in `Appendix I`, does include metrics (the Relationship dimension dimension) to assess the evolution of relationship dynamics. However, in the current study, we observed that the evaluated models struggle with even the foundational aspects of lifelong social interactions.
> >
> > Specifically, the naive models exhibit low believability scores due to characteristic inconsistencies, and even with the memory module, their goal completion scores do not improve significantly in the harder scenarios. These **limitations prevent the models from achieving more complex social intelligence behaviours, such as evolving meaningful relationship dynamics.** Consequently, the relationships remain relatively static throughout the interactions, reflecting the current models’ inability to engage in the deeper cognitive and social processes required for such evolution.  We believe that addressing these foundational challenges is a prerequisite for studying higher-level phenomena like relationship dynamics.
> >
> > > **Q2:** How sensitive are the results to the specific prompt used for generating the episode summaries in the advanced memory module? Have you considered using other metrics beyond BEL and GOAL to evaluate social intelligence, such as measures of empathy, cooperation, or persuasion?
> >
> > **R6:** As detailed in our response to W3, we have conducted an ablation study to examine how different components of the summary in the advanced memory module impact model performance. We hope this addresses the reviewer’s concerns, and we would be happy to provide further details if needed.
> >
> > Regarding the use of additional metrics such as empathy, cooperation, or persuasion, we believe these aspects are implicitly captured under the Goal Completion (GOAL) metric. Strategies like persuasion and cooperation are instrumental in helping agents achieve their goals and thus contribute directly to their GOAL scores. Our evaluation framework focuses on the outcomes of interactions—how effectively agents achieve their goals and maintain consistency with their persona (as captured by BEL and GOAL)—rather than the specific techniques employed during interactions. This outcome-based approach aligns with the objectives of our study, which seeks to evaluate the overarching effectiveness of social intelligence in agents rather than dissecting specific strategies.

---

> > > ### Author Response · Authors · 2024-11-28
> > >
> > > > **Q3:** What are the potential ethical implications of developing LLMs with improved social intelligence, particularly in the context of long-term interactions? The paper touches upon this, but more elaboration would be beneficial. How might the findings of this work inform the design of more socially intelligent LLM agents and their integration into real-world applications?
> > >
> > > **R7:** Thank you for raising the important question regarding the ethical implications of developing LLMs with improved social intelligence in long-term interactions and how our findings may inform real-world applications. We have reflected on these concerns and elaborate below on key ethical aspects.
> > >
> > > Improving the social intelligence of LLMs, particularly in the context of long-term interactions, raises significant ethical considerations, including anthropomorphism, manipulation, and privacy. As discussed in the paper, attributing human traits to AI systems can lead to unrealistic expectations and potentially harmful manipulation [3,4]. Users may form emotional connections or overestimate the system's cognitive and social capabilities, as has been observed in human-technology interactions with conversational agents [2]. These risks are amplified in long-term interactions, where the AI’s perceived relational presence may inadvertently foster trust and reliance, making users more vulnerable to social engineering or exploitation [1].
> > >
> > > To mitigate these risks, LIFELONG-SOTOPIA deliberately avoids the creation of consistent, human-like personalities. Instead, agents are designed to role-play varying characters across scenarios. This approach reduces the risk of anthropomorphism and its associated consequences by preventing the development of consistent, human-like personas.
> > >
> > > **Broader Ethical Considerations:**
> > > 1. *Manipulation and Influence*: Enhanced social intelligence may empower agents to use persuasive techniques that, while effective, could be ethically questionable. For example, agents might exploit contextual cues or emotional states to influence user behavior in ways that may not align with their best interests. As highlighted in prior research [2], addressing the manipulative potential of agentic AI is crucial to ensure ethical deployment.
> > >
> > > 2. *Privacy and Data Ethics*: Long-term interactions require the collection and utilization of user data, raising critical concerns about data security, informed consent, and potential misuse. As noted in prior work [1], transparency regarding data usage and providing users with control over what is collected and retained are fundamental to ethical design.
> > >
> > > **Implications of our findings:**
> > >
> > > The findings in our paper emphasize the gap between current models and human capabilities in leveraging memory and context for social intelligence. Through outcome-oriented evaluations, such as the BEL and GOAL dimensions, we demonstrate specific areas where current models fall short. By providing a robust framework for evaluating social intelligence in LLMs, our work offers a foundation for guiding the design of socially intelligent agents.
> > >
> > > > **Q4:** Would a curriculum learning approach, where agents are gradually exposed to more complex social scenarios, lead to improved performance in LIFELONG-SOTOPIA?
> > >
> > > **R8:** The suggestion to use a curriculum learning approach, where agents are gradually exposed to increasingly complex social scenarios, is indeed intriguing and offers a promising direction for future research. However, the primary focus of this paper is not on training improved agents but rather on benchmarking the performance of existing models in LIFELONG-SOTOPIA. By establishing a robust evaluation framework and identifying the current limitations of these models, we aim to provide a foundation for future work, including approaches like curriculum learning.
> > >
> > > > **Q5:** Could the evaluation be further improved by incorporating a metric that considers the efficiency with which the agents achieve their goals (e.g., number of turns required)? Humans may prioritize efficiency in certain social situations.
> > >
> > > **R9:** In our current framework, agents are already constrained to a maximum number of turns (set to 20 by default) per scenario to achieve their goals. This implicitly penalizes agents that fail to prioritize efficiency, as they are unable to accomplish their objectives within the allotted turn limit.

---

> > > > ### Author Response · Authors · 2024-11-28
> > > >
> > > > References:
> > > >
> > > > [1] Bailey, J. O., Patel, B., & Gurari, D. (2021). A Perspective on Building Ethical Datasets for Children's Conversational Agents. Frontiers in artificial intelligence, 4, 637532. https://doi.org/10.3389/frai.2021.637532
> > > >
> > > > [2] Alberts, L., Keeling, G., & McCroskery, A. (2024). What makes for a'good'social actor? Using respect as a lens to evaluate interactions with language agents. arXiv preprint arXiv:2401.09082.
> > > >
> > > > [3] Deshpande, A., Rajpurohit, T., Narasimhan, K., & Kalyan, A. (2023). Anthropomorphization of AI: opportunities and risks. arXiv preprint arXiv:2305.14784.
> > > >
> > > > [4] Shanahan, M., McDonell, K., & Reynolds, L. (2023). Role play with large language models. Nature, 623(7987), 493-498.
> > > >
> > > > [5] Zhou, X., Zhu, H., Mathur, L., Zhang, R., Yu, H., Qi, Z., ... & Sap, M. (2023). Sotopia: Interactive evaluation for social intelligence in language agents. arXiv preprint arXiv:2310.11667.

---

### Meta-Review · Area_Chair_WLdC · 2024-12-19

**Metareview:**

The paper presents an extension to Sotopia - a LLM-based social simulation to create more socially intelligent agents - to much longer conversations. From the reviewer discussions, it appears that the main concerns revolve around: (1) the fragility of model based evaluations (see review 1E4s) which form the backbone metrics of this work; and (2) the utility of a proposed benchmark with metrics where it is very difficult to outperform a naive closed API based solution (see Fig 4 and review J9sp). This along with other flaws such as not being well situated with the literature of life long learning agents along with a lack of ablations on the agents tested means that this paper is not suitable for publication without significant revisions.

**Additional Comments On Reviewer Discussion:**

I am discounting review fE87 as it is fairly minimal and the confidence relatively low compared to the other reviews. Overall, it does not appear that the authors have sufficiently addressed the reviewers' concerns - it appears that reviewer J9sp had additional questions that were not responded to.

---

### Decision · Program_Chairs · 2025-01-22

Reject